# GROUP FAIRNESS UNDER DISTRIBUTION SHIFTS: ANALYSIS AND ROBUST POST-PROCESSING

## ABSTRACT

Group fairness, as a statistical notion, is sensitive to distribution shifts, which may invalidate the fairness guarantees of classifiers trained with non-robust algorithms. In this work, we analyze randomized fair classifiers and derive upper bounds on fairness violation and excess risk under distribution shift, decomposed into *covariate shift*, and *concept shift*—changes in the distribution of group labels (and other variables considered by the fairness criterion) conditioned on the input. Our bounds are general and apply to both multi-class and attribute-blind settings; notably, we show that attribute-blind classifiers incur an additional dependency on the fairness tolerance in their excess risk, suggesting the robustness benefits of attribute awareness. Next, we propose a robust post-processing algorithm that learns fair classifiers with respect to an uncertainty set constructed by modeling the potential covariate and concept shifts, aligning with the decomposition in our analysis. We evaluate our algorithm under geographic shifts in the ACSIncome dataset, demonstrating improved fairness on unseen regions, with additional evaluations performed under noisy group labels and worst-case covariate shifts.[1]

## 1 INTRODUCTION

Prediction models trained on past data using machine learning techniques are known to exhibit and propagate historical social biases, resulting in disparate impact on or treatment of different demographic groups, e.g., with respect to sex or race (Bolukbasi et al., 2016; Angwin et al., 2016; Buolamwini & Gebru, 2018). To quantify these impacts and assess the unfairness of models, the literature has introduced notions of *group fairness* that examine disparities in the statistics of model outputs across groups (Barocas et al., 2023): *statistical parity* requires equalized group-conditional output distributions (Calders et al., 2009), while *equal opportunity* asks for equalized true positive rates (Hardt et al., 2016). A variety of fair algorithms have since been proposed to satisfy group fairness, which can be categorized by the stage of the training pipeline at which mitigation occurs: preprocessing cleans the data to remove biased associations (Kamiran & Calders, 2012; Calmon et al., 2017), in-processing incorporates fairness constraints into the training objective (Zemel et al., 2013; Agarwal et al., 2018), and post-processing applies post-hoc adjustments to enforce fairness (Menon & Williamson, 2018; Chen et al., 2024; Xian et al., 2023; Xian & Zhao, 2024).

However, group fairness, as a statistical notion, is sensitive to shifts or perturbations in the underlying data distribution, which can arise from changing environments or (adversarial) noise in the training data (Barrainkua et al., 2025). This means that the fairness guarantees of a fair classifier may no longer hold when it is deployed on a distribution that differs from the one it was trained on. Empirically, Ding et al. (2021) consider *geographic shift* and show that an income predictor trained to be fair on one region violates fairness on other regions. We revisit and reproduce this experiment in Fig. 1, where we train a fair classifier on California (CA) data and evaluate it on 26 other states. It is observed that its unfairness (violation of *equalized odds*) increases with the distribution shift from CA, and similarly the excess risk—i.e., *at the same achieved level of fairness on the test distribution, how much worse is the test accuracy of a fair classifier trained on CA compared to one trained directly on the test?* As distribution shifts are prevalent in real-world applications, this brittleness of fair classifiers necessitates further study of the effects of distribution shift and the development of robust algorithms for fair classification.

---

[1]Code will be immediately released after the anonymity period.

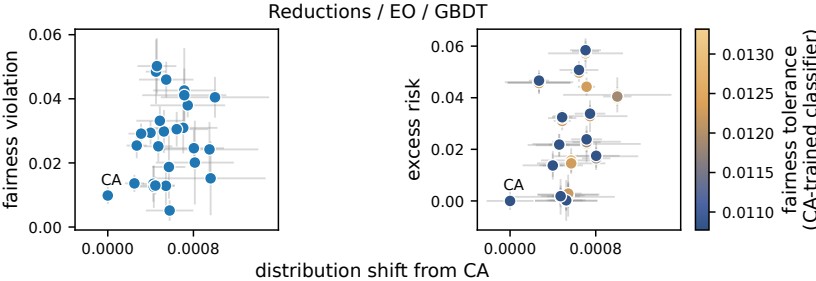

Figure 1: Fair classifiers trained on CA data using the Reductions algorithm of Agarwal et al. (2018) under varying tolerance levels, and evaluated on 26 other states. The $x$-axis measures distribution shift from CA according to Eq. (1) using *maximum mean discrepancy* (Gretton et al., 2012). **Left:** Minimum fairness violation achieved on each state by any CA-trained classifier. **Right:** Excess risk of CA-trained classifiers, measured as the accuracy gap relative to classifiers trained directly on each state with comparable fairness levels (within $0.1\times$); the fairness tolerance of the CA-trained is indicated by color. See Figs. 9 and 10 for results under other criteria and for classifiers trained via LinearPost (Xian & Zhao, 2024).

**Our Contributions.** This work considers randomized fair classifiers under general distribution shifts and studies fairness criteria covering statistical parity, equal opportunity, and equalized odds:

- In Section 3, we analyze and bound the fairness violation in terms of the *covariate* and *concept shift* in the joint distribution of input $X$ and fairness-relevant variables (e.g., $A$ for statistical parity, $(A, Y)$ for equal opportunity and equalized odds). The excess risk bound also includes a term for the shift in $(X, Y)$ distribution, and has a dependency on the fairness tolerance: excess risk can be greater at higher levels of fairness. However, *attribute-aware* classifiers do not exhibit this dependency for certain criteria, suggesting that attribute awareness can benefit robustness.

- In Section 4, we present a post-processing-based algorithm for learning randomized fair classifiers that are robust to distribution shifts. It iterates between finding a (worst-case) perturbation in an *uncertainty set* that maximizes the fairness violation, and learning a classifier that satisfies fairness with respect to all previously found perturbations. We extend the post-processing algorithm of Xian & Zhao (2024) to achieve fairness on multiple distributions, and define the uncertainty set in terms of the potential covariate and concept shifts.

- In Section 5, we evaluate our robust algorithm under geographic shifts in the ACSIncome dataset (Ding et al., 2021), demonstrating improved fairness on unseen regions—albeit, as expected, at a performance cost on the source (training) distribution. We also conduct evaluations in Appendices F and G under noisy group labels and worst-case covariate shift, in the construction of the uncertainty set, respectively.

## 2 PRELIMINARIES

A classification problem is defined by a joint distribution $p$ over features $X \in \mathcal{X}$ (a metric space equipped with distance $d$), class labels $Y \in \mathcal{Y} = \{0, \ldots, K-1\}$, sensitive attributes $A \in \mathcal{A} = \{0, \ldots, G-1\}$ representing demographic groups, and additional variables $Z \in \mathcal{Z}$ relevant to the fairness criterion ($\mathcal{Z} = \emptyset$ for statistical parity, and $Z = Y$ for equal opportunity and equalized odds).

The goal is to learn *randomized classifiers* that satisfy notions of group fairness across the $G$ groups: given an input $x$, the predicted label $\widehat{Y}|X = x$ is sampled from the distribution $\mathrm{multinomial}(h(x))$, where $h : \mathcal{X} \to \Delta(\mathcal{Y})$ is a probabilistic predictor that maps each input to a distribution over class labels. Since $h$ determines the distribution over class assignments, we will refer to the randomized classifier by the function $h$ itself. Given a loss function $\ell : \mathcal{Y} \times \mathcal{Y} \to [0, \infty)$, the risk of a randomized classifier is defined as $R_p(h) = \mathbb{E}[\ell(Y, \widehat{Y})]$, where the expectation is taken over both the data distribution $p$ and the randomness of the classifier.

When the sensitive attribute $A$ is explicitly included as part of the input features $X$, the setting is called *attribute-aware*, as the classifier $\widehat{Y}$ can directly leverage this information. Otherwise, the setting is referred to as *attribute-blind*.

## 2.1 GROUP FAIRNESS DEFINITIONS

We consider group fairness criteria that can be expressed in terms of pairwise differences of first-order conditional moments of the class outputs:

**Definition 2.1.** A (approximate) fairness criterion is defined by a sensitive attribute $A$, a (categorical) event $Z \in \mathcal{Z}$ that does not depend on the classifier $\widehat{Y} \sim \mathrm{multinomial}(h(X))$, and a collection of output class-event pairs specifying the fairness constraints: $\mathcal{C} = \{(y_1, z_1), \ldots, (y_C, z_C)\}$, with $z_c \in \mathcal{Z}$. For a tolerance level $\alpha \in [0, 1]$, we require the fairness violation $V_p(h) \leq \alpha$, defined as

$$V_p(h) = \max_{a,a' \in \mathcal{A}, c \in \{1,\ldots,C\}} |\mathbb{P}(\widehat{Y} = y_c \mid A = a, Z = z_c) - \mathbb{P}(\widehat{Y} = y_c \mid A = a', Z = z_c)|.$$

Here, $\mathbb{P}$ is taken with respect to both the underlying distribution $p$ and the randomness of $\widehat{Y}$. We omit the subscript $p$ when the distribution is clear from context.

Definition 2.1 encompasses common group fairness criteria such as statistical parity, equal opportunity, and equalized odds, but not accuracy parity (Zafar et al., 2017; Zhao & Gordon, 2022) or predictive rate parity (Chouldechova, 2017). A shared property of all criteria of this form is that the constant classifier is always exactly fair ($\alpha = 0$) under any distribution:

**Fact 2.2.** *For the classifier $h(x) = (1, 0, \ldots, 0)$ that always outputs class 0, for every distribution $p$, we have $V_p(h) = 0$.*

Below we recall the fairness criteria that are the focus of our discussions:

- **Statistical Parity** (SP; Calders et al., 2009)**.** Requires the output distributions to be (approximately) equalized across all groups $a \in \mathcal{A}$:

$$V^{\mathrm{SP}} = \max_{a,a' \in \mathcal{A}, k \in \mathcal{Y}} \left|\mathbb{P}(\widehat{Y} = k \mid A = a) - \mathbb{P}(\widehat{Y} = k \mid A = a')\right|.$$

- **Equal Opportunity** (EOpp; Hardt et al., 2016)**.** Defined for binary classification ($K = 2$), assuming class 1 is the more desirable outcome; requires the true positive rate to be equalized:

$$V^{\mathrm{EOpp}} = \max_{a,a' \in \mathcal{A}} \left|\mathbb{P}(\widehat{Y} = 1 \mid A = a, Y = 1) - \mathbb{P}(\widehat{Y} = 1 \mid A = a', Y = 1)\right|.$$

- **Equalized Odds** (EO; Hardt et al., 2016)**.** Can be considered a stricter version of EOpp. It requires all types of classification error to be balanced across groups:

$$V^{\mathrm{EO}} = \max_{a,a' \in \mathcal{A}, j,k \in \mathcal{Y}} \left|\mathbb{P}(\widehat{Y} = k \mid A = a, Y = j) - \mathbb{P}(\widehat{Y} = k \mid A = a', Y = j)\right|.$$

## 2.2 DISTRIBUTION SHIFTS

We are concerned with the robustness of randomized fair classifiers under distribution shifts. The shift from a training distribution $p$ to a test distribution $q$ can be arbitrary, but specific types of shifts have received special attention in the literature (see surveys by Kouw & Loog (2019) and Farahani et al. (2021)). Our main focus is on fairness guarantees, so we consider the shift in $(X, A, Z)$—the input features and variables relevant to the fairness criterion:

- **Covariate Shift.** Decomposing the joint distribution $p_{X,A,Z}$ into $p_X \cdot p_{A,Z|X}$, this model assumes that only the marginal distribution of features $X$ differs between $p$ and $q$. That is, $p_{A,Z|X=x} = q_{A,Z|X=x}$ for all $x \in \mathcal{X}$, while $p_X \neq q_X$. For transfer learning to be feasible, it is often assumed that the density ratio (also called the importance weight) is bounded $q_X(x)/p_X(x) \leq \gamma$ for all $x$.
- **Concept Shift.** This is the opposite of covariate shift: it assumes $p_X = q_X$, while $p_{A,Z|X=x} \neq q_{A,Z|X=x}$. One example is learning under noisy group labels (Wang et al., 2020).
- **Prior Shift.** By decomposing the joint as $p_{A,Z} \cdot p_{X|A,Z}$, this model assumes $p_{X|A=a,Z=z} = q_{X|A=a,Z=z}$ for all $a, z$, while the marginal distribution of $(A, Z)$ differs between $p$ and $q$.

Distribution shifts can be quantified using probability metrics (Zhao et al., 2022), such as $f$-divergences and integral probability metrics. We use metrics from the latter family:

**Definition 2.3** (Dudley Metric). Let $\mathcal{X}$ be a metric space with distance $d$, and let $\mathrm{Lip}(f)$ denote the Lipschitz constant of a function $f : \mathcal{X} \to \mathbb{R}$, that is, $|f(x) - f(x')| \leq \mathrm{Lip}(f)\, d(x, x')$. We define the Dudley probability metric with parameters $B, L$ as

$$D_{B,L}(p, q) = \sup_{f:\mathcal{X}\to[0,B], \mathrm{Lip}(f)\leq L} |\textstyle\int_{\mathcal{X}} f(x) \cdot (p(x) - q(x))\, \mathrm{d}x|.$$

For any $L \leq L'$, the witness function class satisfies $\{f : \mathrm{Lip}(f) \leq L\} \subseteq \{f : \mathrm{Lip}(f) \leq L'\}$, so $D_{B,L} \leq D_{B,L'}$. Also note that the total variation distance is $D_{\mathrm{TV}} = D_{1,\infty} \geq D_{B,L}$ for all $L$. The Dudley metric is also related to the Wasserstein-1 distance via its dual formulation, with the added constraint that $f$ is bounded, so $2\, D_{B,L} \leq L\, W_1$.

## 3 ROBUSTNESS OF RANDOMIZED FAIR CLASSIFIERS

To begin, we analyze the fairness violation and excess risk of randomized fair classifiers under distribution shifts. The bounds are general across problem settings, such as multi-class and attribute-blind, and apply to all fairness criteria defined in Definition 2.1, including SP, EOpp, and EO.

We are primarily interested in the Bayes-optimal randomized fair classifier, defined as $\bar{h} \in \arg\min_{h:V(h)\leq\alpha} R(h)$ for a given fairness tolerance $\alpha$. Our analyses, however, apply to a more refined class of Lipschitz-constrained optimal classifiers that are smooth in their probabilistic predictions with respect to the input $x$, $\bar{h}_L \in \arg\min_{h:V(h)\leq\alpha, \mathrm{Lip}(h)\leq L} R(h)$, where $\mathrm{Lip}(h) \leq L$ means that $|h(x)_k - h(x')_k| \leq L\, d(x, x')$ for all $k \in \mathcal{Y}$. Note that $\bar{h}_\infty$ corresponds to the Bayes-optimal fair classifier.

### 3.1 FAIRNESS VIOLATION

We first bound the fairness violation of a classifier $h$ on the test distribution $q$ by its violation on the source distribution $p$, plus a term on the distributional shift between $p, q$, which we decompose into covariate and concept shift components.

**Theorem 3.1.** *Let $p, q$ be two distributions. Let $h : \mathcal{X} \to \Delta(\mathcal{Y})$ be a Lipschitz randomized classifier with $\mathrm{Lip}(h) \leq L$, then its fairness violation on $q$ satisfies*

$$V_q(h) \leq V_p(h) + 2\max_{a\in\mathcal{A}, z\in\mathcal{Z}} D_{1,L}(p_{X|A=a,Z=z}, q_{X|A=a,Z=z}). \tag{1}$$

*Moreover, if $L' \geq \mathrm{Lip}(x \mapsto q_{A,Z|X=x}(a, z))$ for all $a, z$ (small $L'$ means changes to the conditional probabilities of $(A, Z)$ are smooth), then*

$$V_q(h) \leq V_p(h) + 4\max_{\substack{a\in\mathcal{A} \\ z\in\mathcal{Z}}} \frac{1}{p_{A,Z}(a,z)}\Big(\underbrace{D_{1,(L+1)L'}(p_X, q_X)}_{\text{covariate shift}} + \underbrace{\mathbb{E}_{X\sim p_X}[|p_{A,Z|X}(a,z) - q_{A,Z|X}(a,z)|]}_{\text{concept shift}}\Big).$$

Bounds similar to the first result are shown in (Wang et al., 2020; Agarwal et al., 2025); here, we further decompose it into covariate and concept shifts components: changes in either the marginal distribution of the input $X$ or the conditional distribution of $(A, Z)$ can affect the fairness guarantees established under the source distribution $p$. The bounds yield two insights into robustness. (1) *Smooth randomized classifiers are more robust.* Since $D_{1,L} \leq D_{1,L'}$ for all $L \leq L'$, the bound implies that classifiers with smaller Lipschitz constants (i.e., smoother probabilistic predictors) are more robust to distribution shifts. This motivates the use of Lipschitz-constrained training to improve robustness: a related work is Jiang et al. (2023), who apply *sharpness-aware minimization* in the training of fair classifiers. (2) *Prior shift does not affect fairness.* Because the fairness violation depends only on the conditional distributions $p_{X|A,Z}, q_{X|A,Z}$, changes in the marginal distribution of $(A, Z)$ alone do not impact fairness (consistent with related findings from An et al. (2022)).

In Fig. 1 (left), we plot the best EO fairness achieved on other states by fair classifiers trained on CA (under varying tolerances), against the distribution shift from CA. As expected, fairness violation generally increases with the shift.

## 3.2 Excess Risk

Let $\bar{h}_{p,L}$ be a (Lipschitz) randomized fair classifier that is optimal on the source distribution $p$. We bound the excess risk of $\bar{h}_{p,L}$ on the test distribution $q$ relative to its optimal fair classifier, $\bar{h}_{q,L}$.

**Theorem 3.2.** *Let $p, q$ be two distributions with $L' \geq \mathrm{Lip}(x \mapsto q_{Y|X=x}(y))$. Let $L \in [0, \infty]$, $\alpha \in [0, 1]$, and denote by $\bar{h}_{p,L} \in \arg\min_{h:V_p(h) \leq \alpha, \mathrm{Lip}(h) \leq L} R_p(h)$ the optimal fair classifier on $p$, and by $\bar{h}_{q,L}$ that on $q$. Suppose $V_p(\bar{h}_{q,L}) \leq \alpha + \varepsilon$ for some upper bound $\varepsilon$ on the excess violation (see Theorem 3.1), then the excess risk of $\bar{h}_{p,L}$ on $q$ is*

$$R_q(\bar{h}_{p,L}) - R_q(\bar{h}_{q,L}) \leq \|\ell\|_\infty \left( 2 \, D_{1,(L+L')K}(p_X, q_X) + 2 \, \mathbb{E}_{X \sim p_X}[D_{\mathrm{TV}}(p_{Y|X}, q_{Y|X})] + \frac{\varepsilon}{\alpha + \varepsilon} \right),$$

*with the convention that $0/0 = 0$.*

By instantiating $\varepsilon$ via Theorem 3.1, this result bounds the excess risk in terms of the shifts in the joint distribution of $(X, A, Z)$ and $(X, Y)$, which we similarly decompose into the covariate shift and the concept shift in $Y|X$. Notably, the final term in the bound depends on the fairness tolerance $\alpha$: achieving higher fairness on the source distribution $p$ (i.e., using a smaller $\alpha$) can guarantee better worst-case fairness on the test distribution $q$, but potentially at the cost of higher excess risk. As illustrated in Fig. 1 (right), excess risk grows with increasing shift, but its dependence on the fairness tolerance is weak, suggesting that this worst-case effect may not be dominant in practice.

We illustrate the tightness of this $\alpha$-dependency through a worst-case example for attribute-blind statistical parity that matches the upper bound up to a multiplicative factor (Hou & Zhang (2024) showed the same worst-case dependency on $\alpha$, and established matching minimax lower bounds):

*Example* 1. Let $\alpha \in [0, 1]$ and $\varepsilon \in [0, 1 - \alpha]$. Construct distributions $p, q$ over $(X, A, Y)$ as follows: $p_A = q_A$ uniformly over $\mathcal{A} = \{0, 1\}$; $p_X = q_X$ uniformly over $\mathcal{X} = \{0, 1\}$; $Y = X$; $p(X = 0 \mid A = 0) = (1 - \alpha - \varepsilon)/2$ and $p(X = 0 \mid A = 1) = (1 + \alpha + \varepsilon)/2$; $q(X = 0 \mid A = 0) = (1 - \alpha)/2$ and $q(X = 0 \mid A = 1) = (1 + \alpha)/2$. There is no shift in $(X, Y)$, but the shift in $(X, A)$ is $2 \, D_{\mathrm{TV}}(p_{X|A=a}, q_{X|A=a}) = \varepsilon$ for both $a$. Let $\bar{h}_p$ and $\bar{h}_q$ be Bayes-optimal classifiers satisfying $\alpha$-approximate SP on $p$ and $q$, respectively. Then with the 0-1 loss (classification error), the excess risk is $R_q(\bar{h}_p) - R_q(\bar{h}_q) = \varepsilon/2(\alpha + \varepsilon)$.

The dependency on $\alpha$ can be eliminated if the classifier is attribute-aware, and the fairness criterion is SP, or EO under binary classification (results for the binary and exact fairness case ($\alpha = 0$) are established by Agarwal et al. (2025)). *This highlights the robustness benefits of attribute awareness.*

**Corollary 3.3.** *Assume the attribute-aware setting (i.e., $A$ is included in the classifier input). Under the same conditions as in Theorem 3.2, with the bound $\varepsilon$ instantiated via Theorem 3.1, if the fairness criterion is statistical parity, or equalized odds under binary classification ($K = 2$),[2] then*

$$R_q(\bar{h}_{p,L}) - R_q(\bar{h}_{q,L}) \leq \|\ell\|_\infty (2 \, D_{1,(L+L')K}(p_X, q_X) + 2 \, \mathbb{E}_{X \sim p_X}[D_{\mathrm{TV}}(p_{Y|X}, q_{Y|X})] + 2\varepsilon K).$$

## 4 Learning Robust Fair Classifiers

Given a source distribution $p$ (or labeled examples of) available for training, our goal is to learn a fair classifier that may be deployed on test distribution(s) $q$ differing from $p$. We characterize the potential shifts from the source distribution by a collection $\mathcal{Q}$ of distributions, referred to as the *uncertainty set*. The robust fair classification problem is then formulated as:[3]

$$\arg\min_h R_p(h) \quad \text{s.t.} \quad V_p(h) \leq \alpha, \ V_q(h) \leq \alpha, \ \forall q \in \mathcal{Q}.$$

Algorithm 1 describes a *cutting-set* method[4] for solving the robust problem given a fair classification oracle, and a *pessimization* oracle that finds the worst-case $q \in \mathcal{Q}$ where fairness is most violated.

---

[2]The result also applies to equal opportunity via a similar analysis for equalized odds.

[3]Our formulation minimizes only the source risk $R_p(h)$, rather than the worst-case risk over $\mathcal{Q}$, i.e., $\arg\min_{h:V_q(h) \leq \alpha, \forall q \in \{p\} \cup \mathcal{Q}} \max_{q \in \{p\} \cup \mathcal{Q}} R_q(h)$. The latter can be solved via an additional level of optimization; see e.g., the meta-algorithm of (Mandal et al., 2020, Algorithm 1).

[4]Alternative approaches include online learning techniques (Mandal et al., 2020; Ben-Tal et al., 2009), where $\bar{h}$ is optimized using no-regret algorithms.

---

**Algorithm 1** Robust Fair Classification (Cutting-Set Method)

---

**Require:** Fairness criterion $V$, tolerance $\alpha$, distribution $p$, uncertainty set $\mathcal{Q}$, parameters $T, \tau > 0$.
1:  $\bar{h} \leftarrow \arg\min_h R_p(h)$ s.t. $V_p(h) \leq \alpha$
2:  **for** $t \in \{1, \dots, T\}$ **do**
3:      $q_t \leftarrow \arg\max_{q \in \mathcal{Q}} V_q(\bar{h})$                                $\triangleright$ *pessimization*
4:      **break if** $V_{q_t}(\bar{h}) \leq \alpha + \tau$
5:      $\bar{h} \leftarrow \arg\min_h R_p(h)$ s.t. $V_p(h) \leq \alpha, V_{q_1}(h) \leq \alpha, \dots, V_{q_t}(h) \leq \alpha$   $\triangleright$ *optimization*
6:  **return** $\bar{h}$

---

The algorithm is initialized with the fair classifier on $p$, then iterates between finding a violating perturbation $q_t$ and updating $\bar{h}$ to satisfy fairness on $q_t$ and across all previously found perturbations $p, q_1, \dots, q_{t-1}$. It terminates if no violating perturbation is found up to tolerance $\tau$, in which case the returned classifier $\bar{h}$ satisfies $V_q(\bar{h}) \leq \alpha + \tau$ for all $q \in \mathcal{Q}$ (plus possible slack if the pessimization is approximate or estimating from finite samples), or when the iteration limit $T$ is reached.

The key components in Algorithm 1 are the fair classification oracle (Lines 1 and 5), the pessimization oracle (Line 3), and the specification of the uncertainty set $\mathcal{Q}$. In Section 4.1, we instantiate the fair classification oracle by extending the post-processing algorithm of Xian & Zhao (2024) to multiple distributions. In Section 4.2, we construct the uncertainty set $\mathcal{Q}$ by modeling the covariate and concept shifts from $p$, reflecting the decomposition given in Theorem 3.1. If the shifts $q$ are unknown or adversarial, we model them using parameterized models (e.g., neural nets), and perform pessimization approximately by optimizing the parameters of $q$ (via gradient ascent) to maximize the fairness violation $V_q$, with a regularization term to enforce bounded divergence from $p$.

**Convergence of Algorithm 1.**    Assume that the input space $\mathcal{X}$ has finite support, i.e., $\mathrm{supp}(q_X) = \mathrm{supp}(p_X) = \mathcal{X}$ for all $q \in \mathcal{Q}$ and $N \coloneqq |\mathcal{X}| < \infty$ (as is the case when learning from finite samples). Then a randomized classifier $\bar{h}$ can be represented as an $N \times K$ row-stochastic matrix, where the $i$-th row gives the output distribution for the $i$-th input. Moreover, the fairness violation $V$ is 1-Lipschitz in $h$ under the $L^\infty$-distance: $|V(h) - V(h')| \leq \|h - h'\|_\infty$ (see derivation in Eq. (5)).

Then, the analysis of (Mutapcic & Boyd, 2009, Section 5.2) shows that Algorithm 1 terminates in at most $O(\tau^{-NK})$ iterations. The intuition is as follows: each time a violation exceeding $\tau$ is found, the updated $\bar{h}$ must be at least $\tau$ away (in $L^\infty$-distance) from the current $\bar{h}$ to restore fairness due to the Lipschitzness of $V$, effectively removing a $\ell_\infty$-ball of radius $\tau$ from the feasible region (hence the name *cutting-set*). Since only $O(\tau^{-NK})$ such balls can be packed into the space of randomized classifiers where each cell is bounded between $[0, 1]$, the algorithm must terminate within this bound. In practice, however, far fewer iterations are needed: in our experiments, it typically terminates within 5 to 20 iterations.

### 4.1    FAIR CLASSIFICATION VIA POST-PROCESSING

Algorithm 1 requires a fair classification oracle, either for a single distribution (Line 1) or simultaneously across multiple distributions (Line 5). To implement this, we use the post-processing algorithm LinearPost proposed by Xian & Zhao (2024) for the single-distribution setting (reviewed below), and extend it to the multiple-distribution setting.

**Single-Distribution LinearPost.**    LinearPost learns fair classifiers for a single distribution $p$ by fitting a linear classifier on top of the outputs from a predictor $p_{A,Z|X} : \mathcal{X} \to \Delta(\mathcal{A} \times \mathcal{Z})$ for the conditional distribution of $(A, Z)$ given $X$, and a predictor for the *point-wise risk*, $r_p : \mathcal{X} \to [0, \infty)^K$ (hence called a *post-processing* algorithm),

$$r_p(x)_k = \mathbb{E}_p[\ell(Y, k) \mid X = x], \quad \forall x \in \mathcal{X}, \, k \in \mathcal{Y}, \tag{2}$$

which represents the expected loss of assigning class $k$ to input $x$, e.g., for the 0-1 loss, $r_p(x)_k = p(Y \neq k \mid X = x) = 1 - p_{Y|X=x}(k)$; the overall risk is then $R_p(h) = \mathbb{E}_{X \sim p_X}[r_p(X)^\top h(X)]$.

It is based on the following theorem, which shows that if the predictors $p_{A,Z|X}$ and $r_p$ are Bayes-optimal, then the Bayes-optimal (randomized) fair classifier on $p$ is a linear classifier over $(K + G|\mathcal{Z}|)$-dimensional features computed from $(r_p(x), p_{A,Z|X=x})$, under a mild continuity condition.

This condition can be satisfied by randomly perturbing the point-wise risk $r_p$, which is the only source of randomness in the resulting classifier.

**Theorem 4.1** (Xian & Zhao, 2024). *Let $p$ be a distribution and $\alpha \in [0, 1]$. Assume that the push-forward distribution $r_p \sharp p_X$ is continuous. Then an optimal classifier to the single-distribution fair classification problem, $\arg\min_h R_p(h)$ s.t. $V_p(h) \leq \alpha$, is, for some weights $\beta \in \mathbb{R}^{K \times G \times |\mathcal{Z}|}$,*

$$x \mapsto \arg\min_{k \in \mathcal{Y}}(r_p(x)_k + \textstyle\sum_{a \in \mathcal{A}, z \in \mathcal{Z}} \beta_{k,a,z} p_{A,Z|X=x}(a, z)).$$

The weights $\beta$ are obtained from the dual solution of a linear program (LP) that expresses the single-distribution fair classification problem (details are given in Appendix D.1, Eq. (9)).

**Learning Fair Classifier from Samples.** Given labeled examples $(x_i, a_i, z_i, y_i) \sim p$, we apply a "pre-train then post-process" procedure to learn a fair classifier via LinearPost from scratch. We first learn the predictors $\hat{p}_{A,Z|X}$ and $\hat{r}_p$ in the pre-training step, then invoke LinearPost on them to obtain the classifier. The post-processing weights $\beta$ are estimated by solving the fair classification LP mentioned above, except that, in this case, it is formulated using the empirical distribution of the samples, and the learned $\hat{p}_{A,Z|X}, \hat{r}_p$ as proxies in place of the Bayes-optimal $p_{A,Z|X}, r_p$.

To learn the group predictor $\hat{p}_{A,Z|X}$, we fit a probabilistic classifier (e.g., logistic regression) to predict $(A, Z)$ from $X$ (possibly followed by a calibration step), as this can be viewed as an $(A, Z)$ classification task. Similarly, to learn the point-wise risk predictor $\hat{r}_p$, we fit a model for $Y$ given $X$ and transform its output according to the chosen loss $\ell$ (e.g., 0-1 loss as shown above). To ensure good generalization, the training samples should be split into disjoint sets for pre-training and post-processing; note that post-processing does not require labels as it relies on the predictors as proxies.

**Multiple-Distribution LinearPost.** We extend LinearPost to achieve fairness on multiple distributions $p, q_1, \ldots, q_T$ simultaneously by showing that the Bayes-optimal fair classifier in this setting remains a linear post-processing rule, now over $(K + (T+1)G|\mathcal{Z}|)$-dimensional features. These include $r_p(x)$, $p(A, Z \mid X = x)$, and $w_t(x) \cdot q_t(A, Z \mid X = x)$ for each $t$, where $w_t(x) = q_{tX}(x)/p_X(x)$ is the importance weight between $p_X$ and $q_{tX}$. This additional term is natural, as $w_t \cdot p_X = q_X$ and $q_{tA,Z|X}$ together fully specify the joint distribution $q_{tX,A,Z}$.

**Theorem 4.2** (Multiple-Distribution LinearPost). *Let $p, q_1, \ldots, q_T$ be distributions and $\alpha \in [0, 1]$. Assume that the push-forward distributions $r \sharp p_X, r \sharp q_{1X}, \ldots, r \sharp q_{TX}$ are continuous. Then an optimal classifier to the multiple-distribution fair classification problem, $\arg\min_h R_p(h)$ s.t. $V_p(h) \leq \alpha, V_{q_1}(h) \leq \alpha, \ldots, V_{q_T}(h) \leq \alpha$, is, for some weights $\beta \in \mathbb{R}^{(T+1) \times K \times G \times |\mathcal{Z}|}$,*

$$x \mapsto \underset{k \in \mathcal{Y}}{\arg\min} \left( r_p(x)_k + \sum_{a \in \mathcal{A}, z \in \mathcal{Z}} \left( \beta_{0,k,a,z} p_{A,Z|X=x}(a, z) + \sum_{t=1}^{T} \beta_{t,k,a,z} q_{tA,Z|X=x}(a, z) w_t(x) \right) \right).$$

The weights $\beta$ are again obtained from the dual solution of an LP that expresses the multiple-distribution fair classification problem (Eq. (10)). Thus, to learn a classifier that is fair on $q_1, \ldots, q_M$ in addition to $p$, we apply the multiple-distribution variant of LinearPost by providing descriptions of the $q_m$'s in terms of predictors $q_{mA,Z|X}$ and models for computing the importance weights $w_m$.

### 4.2 PARAMETERIZED DISTRIBUTION SHIFT MODELS

As described in Section 4.1, our implementation of the fair classification oracle in Line 5 uses LinearPost, which requires each perturbation $q_1, \ldots, q_T$ to be specified via its conditional distribution $q_{tA,Z|X}$ and its marginal distribution through the importance weight $w_t = q_{tX}/p_X$. Therefore, we model each $q \in \mathcal{Q}$ in the uncertainty set accordingly by a pair of functions: $f_{\text{CS}} : \mathcal{X} \to \Delta(\mathcal{A} \times \mathcal{Z})$ for the concept shift, and $f_{\text{IW}} : \mathcal{X} \to [0, \infty)$ for the covariate shift from the source distribution $p$. This also mirrors the decomposition of fairness violation into concept and covariate shifts as analyzed in Theorem 3.1. Moreover, these functions must generalize beyond the training set at test time, because classifiers produced by LinearPost rely on their outputs to make predictions (i.e., post-processing).

When knowledge about the potential shifts is available, it can be used to directly specify $\mathcal{Q}$. For example, to model covariate shift in the domain adaptation setting where unlabeled data from the test distribution $q$ are available, we let $\mathcal{Q} = \{q\}$ be a singleton set, with the pair $f_{\text{CS}} = q_{A,Z|X} = $

$p_{A,Z|X}$ unchanged and $f_{\text{IW}} = q_X/p_X$ for the covariate shift, which can be estimated from the samples (Shimodaira, 2000; Coston et al., 2019).

When the shifts are unknown or adversarial, we define $\mathcal{Q}$ as a set of bounded perturbations around the source distribution $p$ (Mandal et al., 2020). For example, to model noisy group labels (Wang et al., 2020), we can let $\mathcal{Q} = \{q : \mathbb{E}_{X \sim p_X}[d_{\text{TV}}(q_{A|X}, p_{A|X})] \leq \gamma\}$. In such cases, to model and approximately find the worst-case perturbation, we parameterize the functions $f_{\text{CS}}$ and $f_{\text{IW}}$ that represent $q$ (e.g., neural nets), and optimize their parameters (via gradient ascent) to maximize the fairness violation $V_q$, with regularization terms to control their deviation from $p$ (and also to prevent overfitting).

We derive the regularized objectives for optimizing the worst-case perturbation $q$ below. Recall that $f_{\text{CS}}$ represents $f_{\text{CS}}(x)_{a,z} = q(A = a, Z = z \mid X = x)$ and $f_{\text{IW}}(x) = q(X = x)/p(X = x)$, so by Bayes' rule, we can express the fairness violation (Definition 2.1) in terms of these functions as

$$V_q(h) = \max_{\substack{a,a' \in \mathcal{A} \\ c \in \{1,\dots,C\}}} \left| \mathbb{E}_{X \sim p_X} \left[ h_{y_c}(X) \left( \frac{f_{\text{CS}}(X)_{a,z_c}}{q_{A,Z}(a, z_c)} - \frac{f_{\text{CS}}(X)_{a',z_c}}{q_{A,Z}(a', z_c)} \right) f_{\text{IW}}(X) \right] \right| \tag{3}$$

where $q_{A,Z}(a, z) = \mathbb{E}_{X \sim p_X}[f_{\text{CS}}(X)_{a,z} f_{\text{IW}}(X)]$. We use KL divergence for regularization:

- **Concept Shift Model.** We regularize $f_{\text{CS}}$ by its average KL divergence from $p_{A,Z|X}$ over $X$, with strength $\lambda_{\text{CS}}$. The objective becomes:

$$\max_{f_{\text{CS}}} \left( V_q(h) - \lambda_{\text{CS}} \, \mathbb{E}_{X \sim p_X}[D_{\text{KL}}(p_{A,Z|X}, f_{\text{CS}}(X))] \right)$$
$$= \max_{f_{\text{CS}}} \left( V_q(h) + \lambda_{\text{CS}} \, \mathbb{E}_{X \sim p_X} \left[ \sum_{a,z} p_{A,Z|X}(a, z) \ln f_{\text{CS}}(x)_{a,z} \right] \right).$$

- **Covariate Shift Model.** We regularize $f_{\text{IW}}$ by its KL divergence from 1 (i.e., between $q_X = p_X \cdot f_{\text{IW}}$ and $p_X$), with strength $\lambda_{\text{IW}}$. The objective becomes:

$$\max_{f_{\text{IW}}}(V_q(h) - \lambda_{\text{IW}} D_{\text{KL}}(p_X, p_X f_{\text{IW}}(X))) = \max_{f_{\text{IW}}}(V_q(h) + \lambda_{\text{IW}} \, \mathbb{E}_{X \sim p_X}[\ln f_{\text{IW}}(X)]).$$

In our experiments, we replace the $\max$ in Eq. (3) with a weighted sum using $\text{softmax}$ to improve optimization performance. We parameterize both functions using one-hidden-layer LeakyReLU nets that take the logits of $p_{A,Z|X}$ as input. For example, we define $f_{\text{IW}}(x) = C \exp(g(\ln p_{A,Z|X=x}))$, where $g$ is the neural net and $C$ is a normalization term such that $\sum_{i=1}^{N} f_{\text{IW}}(x_i)p_X(x_i) = 1$ over the training data (recall $p_X = 1/N$ for empirical distributions).

## 5 Experiments for Geographic Shift

We evaluate the robust fair post-processing algorithm of Section 4 under geographic shifts using the ACSIncome dataset (2018 data; Ding et al., 2021). The task is binary classification of whether an individual's annual income exceeds \$50k, with sex as the binary sensitive attribute. The data is partitioned by the individual's home US state or territory (51 regions in total), but we retain only the top 27 with the largest sample sizes. California (CA) is used as the training/source distribution. To quantify the distribution shift from CA, following the first bound in Theorem 3.1, we compute the maximum mean discrepancy (MMD; Gretton et al., 2012) of the input features $X$ conditioned on $A$ for SP and on $(A, Y)$ for EOpp and EO, using a Gaussian kernel with bandwidth 1; for improved statistical power, we average the conditional MMDs rather than taking their max.

We consider SP, EOpp, and EO fairness in the attribute-blind setting, and follow the "pre-train then post-process" procedure from Section 4.1 to obtain fair classifiers via (robust) LinearPost with a gradient-boosted decision tree (GBDT; Ke et al., 2017) as the base prediction model. The dataset is split 60/10/30 for pre-training, post-processing, and testing. We first fit a GBDT to predict $(A, Y)$ from $X$, then apply (robust) LinearPost; the uncertainty set is implemented using the covariate and concept shift models described in Section 4.2, parameterized by one-hidden-layer neural nets with LeakyReLU activation. We also evaluate the Reductions algorithm (Agarwal et al., 2018). The fair algorithms are applied separately for each fairness criterion. Further details and hyperparameters are provided in Appendix E.

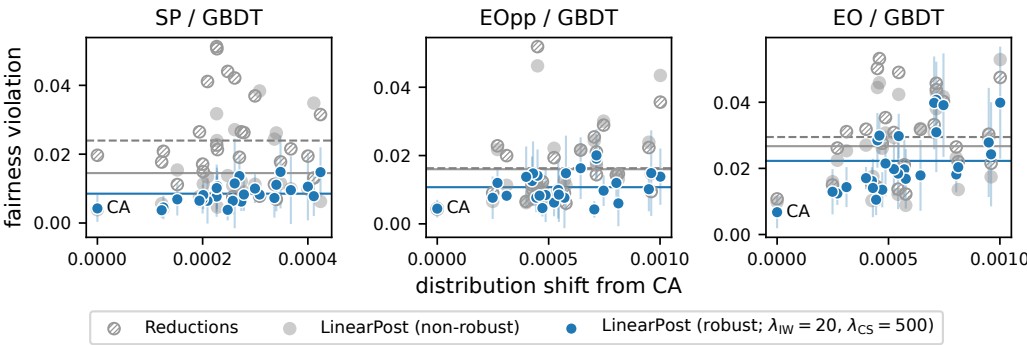

Figure 2: Fairness on each region by Reductions and LinearPost (non-robust and robust with $\lambda_{\text{IW}} = 20$, $\lambda_{\text{CS}} = 500$) trained on CA data, under the tolerance setting that minimizes macro average violation. See Table 1 for the tolerances, average accuracies, and violations (horizontal lines).

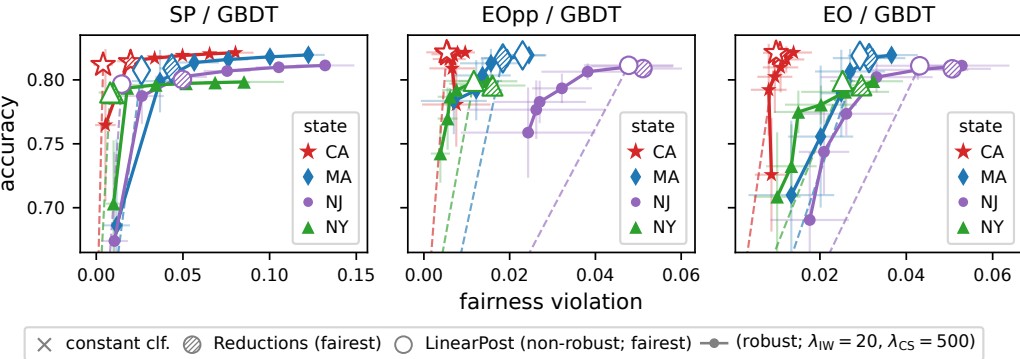

Figure 3: Accuracy-fairness tradeoffs on each region by robust LinearPost ($\lambda_{\text{IW}} = 20$, $\lambda_{\text{CS}} = 500$) trained on CA data. For comparison, we include the fairest Pareto-optimal classifiers from non-robust LinearPost and Reductions, as well as the randomized interpolation between the fairer baseline and the constant 0 classifier (dashed lines).

**Results.** Figure 2 shows the fairness achieved by the classifier from robust LinearPost on all 27 regions (including CA), under the best tolerance setting $\alpha^*$ that minimizes average fairness violation,[5] compared to results from non-robust algorithms. Robust post-processing (under $\alpha^*$) improves fairness both on average and in the worst-performing region, though the improvements are not uniform across all regions. This variation is expected, as the actual distribution shifts may not be fully captured by the perturbation models used to define the uncertainty set—especially since the setting assumes no prior knowledge of the shift; additionally, the optimal $\alpha^*$ and hyperparameters for the perturbation model (e.g., $\lambda_{\text{IW}}$, $\lambda_{\text{CS}}$) may differ across regions. Nonetheless, robust LinearPost's ability to reduce worst-case fairness violation underscores its practical utility.

The fairness improvements come at the cost of reduced accuracy, including on the training distribution. In Fig. 3, we plot the accuracy-fairness tradeoffs achieved by robust LinearPost on the three most violating regions for each fairness criterion, under varying tolerances $\alpha$. For reference, we include the linear randomized interpolation between the fairer baseline and the constant 0 classifier (which is trivially fair). In most cases, (portions of) the Pareto-optimal tradeoff curve of robust LinearPost lies above the interpolation line, indicating that its improvements are non-trivial, except on MO and LA for EOpp fairness, likely because the perturbation models fail to capture the true underlying shifts. We do also observe that fairness does not always improve monotonically as $\alpha$ decreases, and many configurations do not lie on the Pareto front; this may be due to the pessimization step not being performed exactly, as well as variability in the optimization of the perturbation models. It is therefore recommended to validate on the test distribution(s) when selecting hyperparameters and models that effectively improve fairness while maintaining a balance with accuracy.

---

[5]We sweep tolerance settings down to $\alpha = 0.001$, but it may not yield the fairest classifier.

Note on LLM Usage in Paper Writing

The writing of this paper was assisted by OpenAI's GPT model, limited to grammar correction and sentence refinement.

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

## A   RELATED WORK

**Fairness Under Distribution Shifts.**   Our analysis is similar to those in (Wang et al., 2020; Hou & Zhang, 2024; Agarwal et al., 2025). Wang et al. (2020) study fair classifiers under covariate shift and derive bounds on fairness violation in terms of the magnitude of the shift. Agarwal et al. (2025) additionally bound the excess risk of the optimal fair classifier under shifts, but their results are limited to the attribute-aware setting. Hou & Zhang (2024) study the excess risk of the optimal attribute-blind fair classifier, revealing a similar worst-case dependency on the fairness tolerance; while we provide an example that matches this dependency, they establish matching minimax bounds.

In addition, Konstantinov & Lampert (2022) and Blum et al. (2024) analyze fair classifiers under adversarial noise (i.e., worst-case distribution shift), with emphasis on the brittleness of deterministic fair classifiers relative to randomized ones, and Chen et al. (2022) provide fine-grained bounds under covariate and label shifts. Giguere et al. (2022) and Kang et al. (2022) study the problem of certifying fairness guarantees under distribution shifts.

**Robust Fair Algorithms.**   Existing algorithms can be broadly categorized into *domain adaptation* and *generalization* methods (Barrainkua et al., 2025). The former assumes a source and a specific target distribution, with the goal of achieving fairness on the target. The common strategy is to relate the target distribution to the source, via importance weighting, invariant representation learning, or assuming a generative model (e.g., causal graphs), followed by applying standard (non-robust) fair algorithms (Schumann et al., 2019; Coston et al., 2019; Roh et al., 2020; Rezaei et al., 2021; Singh et al., 2021; An et al., 2022; Wu et al., 2022). Generalization methods assume less knowledge about the test distribution(s) and instead define an uncertainty set, often as bounded perturbations around the source distribution. The goal is to ensure fairness under all perturbations in the set, typically using techniques from (distributionally) robust optimization (Wang et al., 2020; Mandal et al., 2020; Jiang et al., 2023; Baharlouei et al., 2024). Our robust fair post-processing algorithm belongs primarily to the latter category but can be adapted to the domain adaptation setting by customizing the uncertainty set based on knowledge of the target distribution.

## B   PROOFS FOR SECTION 3.1

To begin, we derive alternative expressions for the fairness violation in Definition 2.1:

$$V_p(h) = \max_{\substack{a,a' \in \mathcal{A} \\ c \in \{1,\dots,C\}}} \left| \mathbb{P}_p(\widehat{Y} = y_c \mid A = a, Z = z_c) - \mathbb{P}_p(\widehat{Y} = y_c \mid A = a', Z = z_c) \right|.$$

Because the distribution of $\widehat{Y}$ is fully determined by $h$ given $X$, the statistics considered in the group fairness constraint above can be written by Bayes' rule as

$$\mathbb{P}_p(\widehat{Y} = k \mid A = a, Z = z) = \int_{\mathcal{X}} h(x)_k p_{X|a,z}(x)\, \mathrm{d}x = \int_{\mathcal{X}} h(x)_k \frac{p_{A,Z|x}(a,z) p_X(x)}{p_{A,Z}(a,z)}\, \mathrm{d}x. \quad (4)$$

Written in this form, it is easy to show that $V$ is 1-Lipschitz in $h$ in the uniform distance. By Hölder's inequality,

$$\begin{aligned}
|V(h) - V(h')| &= \left| \max_{a,a',z,c} \int_{\mathcal{X}} (h(x)_{y_c} - h'(x)_{y_c}) p_{X|a,z}(x)\, \mathrm{d}x \right| \\
&\le \max_{a,a',z,c} \left( \max_x |h(x)_{y_c} - h'(x)_{y_c}| \right) \int_{\mathcal{X}} p_{X|a,z}(x)\, \mathrm{d}x \\
&= \|h - h'\|_\infty,
\end{aligned} \quad (5)$$

where we defined the $L^\infty$-distance between two (vector-valued) functions $h, h' : \mathcal{X} \to \mathbb{R}^K$ as

$$\|h - h'\|_\infty = \max_{x,k} |h(x)_k - h'(x)_k|.$$

**Lemma B.1.** *Let $p, q$ be two distributions. Let $h : \mathcal{X} \to \Delta(\mathcal{Y})$ be a Lipschitz randomized classifier with $\mathrm{Lip}(h) \le L$, and $\widehat{Y} \sim \mathrm{Multinomial}(h(X))$. Then for any $a \in \mathcal{A}$, $z \in \mathcal{Z}$, and $k \in \mathcal{Y}$, the change in the statistics examined by the group fairness criteria of Definition 2.1 is bounded by*

$$\left| \mathbb{P}_p(\widehat{Y} = k \mid A = a, Z = z) - \mathbb{P}_q(\widehat{Y} = k \mid A = a, Z = z) \right| \le D_{1,L}\big(p_{X|a,z}, q_{X|a,z}\big).$$

*Moreover, if $L' \geq \mathrm{Lip}(x \mapsto q_{A,Z|X=x}(a,z))$ for all $a,z$, then*

$$\left| \mathbb{P}_p(\widehat{Y} = k \mid A = a, Z = z) - \mathbb{P}_q(\widehat{Y} = k \mid A = a, Z = z) \right|$$
$$\leq 2 \big( D_{1,(L+1)L'}(p_X, q_X) + \mathbb{E}_{X \sim p_X} \big| p_{A,Z|X}(a,z) - q_{A,Z|X}(a,z) \big| \big).$$

*Proof.* For the first bound, we use the first alternative form of Eq. (4). By the definition of the Dudley metric (Definition 2.3) and the assumption that $\mathrm{Lip}(h) \leq L$,

$$\left| \int_{\mathcal{X}} h(x)_k \big( p_{X|a,z}(x) - q_{X|a,z}(x) \big) \, \mathrm{d}x \right| \leq D_{1,L} \big( p_{X|a,z}, q_{X|a,z} \big).$$

For the second bound, we use the second form in Eq. (4).

$$\left| \int_{\mathcal{X}} h(x)_k \left( \frac{p_{A,Z|x}(a,z)p_X(x)}{p_{A,Z}(a,z)} - \frac{q_{A,Z|x}(a,z)q_X(x)}{q_{A,Z}(a,z)} \right) \mathrm{d}x \right|$$
$$\leq \left| \int_{\mathcal{X}} h(x)_k \left( \frac{p_{A,Z|x}(a,z)p_X(x)}{p_{A,Z}(a,z)} - \frac{q_{A,Z|x}(a,z)p_X(x)}{p_{A,Z}(a,z)} \right) \mathrm{d}x \right|$$
$$+ \left| \int_{\mathcal{X}} h(x)_k \left( \frac{q_{A,Z|x}(a,z)p_X(x)}{p_{A,Z}(a,z)} - \frac{q_{A,Z|x}(a,z)q_X(x)}{p_{A,Z}(a,z)} \right) \mathrm{d}x \right|$$
$$+ \left| \int_{\mathcal{X}} h(x)_k \left( \frac{q_{A,Z|x}(a,z)q_X(x)}{p_{A,Z}(a,z)} - \frac{q_{A,Z|x}(a,z)q_X(x)}{q_{A,Z}(a,z)} \right) \mathrm{d}x \right|,$$
$$\leq \frac{1}{p_{A,Z}(a,z)} \big( \mathbb{E}_{X \sim p_X} \big| p_{A,Z|X}(a,z) - q_{A,Z|X}(a,z) \big| + D_{1,LL'}(p_X, q_X) \big) \quad (6)$$
$$+ \left| \frac{1}{p_{A,Z}(a,z)} - \frac{1}{q_{A,Z}(a,z)} \right| \left| \int_{\mathcal{X}} h(x)_k q_{X,A,Z}(x,a,z) \, \mathrm{d}x \right|$$

by triangle inequality, and the assumption that $\mathrm{Lip}(q_{A,Z|\cdot}) \leq L'$; continuing with the last term,

$$\left| \frac{1}{p_{A,Z}(a,z)} - \frac{1}{q_{A,Z}(a,z)} \right| \left| \int_{\mathcal{X}} h(x)_k q_{X,A,Z}(x,a,z) \, \mathrm{d}x \right|$$
$$\leq \left| \frac{q_{A,Z}(a,z)}{p_{A,Z}(a,z)} - 1 \right| = \frac{1}{p_{A,Z}(a,z)} |q_{A,Z}(a,z) - p_{A,Z}(a,z)|, \quad (7)$$

where

$$|q_{A,Z}(a,z) - p_{A,Z}(a,z)|$$
$$= \left| \int_{\mathcal{X}} \big( q_{A,Z|x}(a,z)q_X(x) - p_{A,Z|x}(a,z)p_X(x) \big) \, \mathrm{d}x \right|$$
$$\leq \left| \int_{\mathcal{X}} \big( q_{A,Z|x}(a,z) - p_{A,Z|x}(a,z) \big) p_X(x) \, \mathrm{d}x \right| + \left| \int_{\mathcal{X}} q_{A,Z|x}(a,z)(q_X(x) - p_X(x)) \, \mathrm{d}x \right|$$
$$\leq \mathbb{E}_{X \sim p_X} \big| p_{A,Z|X}(a,z) - q_{A,Z|X}(a,z) \big| + D_{1,L'}(p_X, q_X).$$

Combing this with Eqs. (6) and (7) gives the result in the lemma statement. $\qquad \square$

*Proof of Theorem 3.1.* For the first bound, we use the first alternative form of Eq. (4). By triangle inequality, for any $a, a' \in \mathcal{A}$, $z \in \mathcal{Z}$, and $k \in \mathcal{Y}$,

$$\left| \int_{\mathcal{X}} h(x)_k \big( q_{X|a,z}(x) - q_{X|a',z}(x) \big) \, \mathrm{d}x \right|$$
$$\leq \left| \int_{\mathcal{X}} h(x)_k \big( p_{X|a,z}(x) - p_{X|a',z}(x) \big) \, \mathrm{d}x \right|$$
$$+ \left| \int_{\mathcal{X}} h(x)_k \big( p_{X|a,z}(x) - q_{X|a,z}(x) \big) \, \mathrm{d}x \right| + \left| \int_{\mathcal{X}} h(x)_k \big( p_{X|a',z}(x) - q_{X|a',z}(x) \big) \, \mathrm{d}x \right|$$
$$\leq \alpha + D_{1,L} \big( p_{X|a,z}, q_{X|a,z} \big) + D_{1,L} \big( p_{X|a',z}, q_{X|a',z} \big);$$

the last line is from the assumption that $V_p(h) \leq \alpha$ and by Lemma B.1. Then $V_q(h) \leq \alpha + 2\max_{a,z\in\mathcal{Z}} D_{1,L}(p_{X|a,z}, q_{X|a,z})$ by taking the max of the above over $a \in \mathcal{A}$ and $z \in \mathcal{Z}$.

For the second bound, we use the second form of Eq. (4). Again, by triangle inequality,

$$
\left| \int_{\mathcal{X}} h(x)_k \left( \frac{q_{A,Z|x}(a,z)q_X(x)}{q_{A,Z}(a,z)} - \frac{q_{A,Z|x}(a',z)q_X(x)}{q_{A,Z}(a',z)} \right) \mathrm{d}x \right|
$$
$$
\leq \left| \int_{\mathcal{X}} h(x)_k \left( \frac{p_{A,Z|x}(a,z)p_X(x)}{p_{A,Z}(a,z)} - \frac{p_{A,Z|x}(a',z)p_X(x)}{p_{A,Z}(a',z)} \right) \mathrm{d}x \right|
$$
$$
+ \left| \int_{\mathcal{X}} h(x)_k \left( \frac{p_{A,Z|x}(a,z)p_X(x)}{p_{A,Z}(a,z)} - \frac{q_{A,Z|x}(a,z)q_X(x)}{q_{A,Z}(a,z)} \right) \mathrm{d}x \right|
$$
$$
+ \left| \int_{\mathcal{X}} h(x)_k \left( \frac{p_{A,Z|x}(a',z)p_X(x)}{p_{A,Z}(a',z)} - \frac{q_{A,Z|x}(a',z)q_X(x)}{q_{A,Z}(a',z)} \right) \mathrm{d}x \right|,
$$

where the term on the second line is no more than $\alpha$, and the other two terms are bounded using the second result of Lemma B.1. □

## C    PROOFS FOR SECTION 3.2

We first provide the proofs to Theorem 3.2 and Corollary 3.3, then derive the results in Example 1. To simplify notation, we drop the Lipschitz constant $L$ in $\bar{h}_{p,L}, \bar{h}_{q,L}$ in the proofs.

*Proof of Theorem 3.2.* We begin with the following decomposition of the risk:

$$
R_q(\bar{h}_p) - R_q(\bar{h}_q)
$$
$$
= (R_q(\bar{h}_p) - R_p(\bar{h}_p)) + (R_p(\bar{h}_p) - R_p(\bar{h}_q)) + (R_p(\bar{h}_q) - R_q(\bar{h}_q)).
$$

For the first term (and similarly the last),

$$
R_q(\bar{h}_p) - R_p(\bar{h}_p)
$$
$$
= \int_{\mathcal{X}\times\mathcal{Y}} \sum_{k\in\mathcal{Y}} \ell(y,k)\bar{h}_p(x)_k q_{X,Y}(x,y)\,\mathrm{d}xy - \int_{\mathcal{X}\times\mathcal{Y}} \sum_{k\in\mathcal{Y}} \ell(y,k)\bar{h}_p(x)_k p_{X,Y}(x,y)\,\mathrm{d}xy
$$
$$
= \int_{\mathcal{X}\times\mathcal{Y}} \left( \sum_{k\in\mathcal{Y}} \ell(y,k)\bar{h}_p(x)_k \right) (q_{X,Y}(x,y) - p_{X,Y}(x,y))\,\mathrm{d}xy
$$
$$
=: \int_{\mathcal{X}\times\mathcal{Y}} g(x,y)(q_{X,Y}(x,y) - p_{X,Y}(x,y))\,\mathrm{d}xy
$$
$$
= \int_{\mathcal{X}} \sum_{y\in\mathcal{Y}} g(x,y)p_X(x)\big(q_{Y|X=x}(y) - p_{Y|X=x}(y)\big)\,\mathrm{d}x
$$
$$
+ \|\ell\|_\infty \int_{\mathcal{X}} \sum_{y\in\mathcal{Y}} \frac{g(x,y)}{\|\ell\|_\infty} q_{Y|X=x}(y)(q_X(x) - p_X(x))\,\mathrm{d}x
$$
$$
\leq \|\ell\|_\infty \mathbb{E}_{X\sim p_X}[D_{\mathrm{TV}}(p_{Y|X}, q_{Y|X})] + \|\ell\|_\infty D_{1,(L+L')K}(p_X, q_X)
$$

where we defined $g$ to be the expected risk incurred by $\bar{h}_p$ on each $(x,y)$ pair from the underlying distribution; the last line is because $x \mapsto \sum_{y\in\mathcal{Y}} g(x,y)q_{Y|X=x}(y)/\|\ell\|_\infty \in [0,1]$ and is $(L+L')K$-

Lipschitz:

$$\left| \sum_{y\in\mathcal{Y}} g(x,y)q_{Y|X=x}(y) - \sum_{y\in\mathcal{Y}} g(x',y)q_{Y|X=x'}(y) \right|$$

$$\leq \sum_{y\in\mathcal{Y}} q_{Y|X=x}(y)|g(x,y)-g(x',y)| + \sum_{y\in\mathcal{Y}} g(x',y)\big|q_{Y|X=x}(y)-q_{Y|X=x'}(y)\big|$$

$$\leq \sum_{y\in\mathcal{Y}} |g(x,y)-g(x',y)| + \|\ell\|_\infty \sum_{y\in\mathcal{Y}} \big|q_{Y|X=x}(y)-q_{Y|X=x'}(y)\big|$$

$$\leq \sum_{y\in\mathcal{Y}} \left| \sum_{k\in\mathcal{Y}} \ell(y,k)\big(\bar{h}_p(x)_k - \bar{h}_p(x')_k\big) \right| + \|\ell\|_\infty L'K\, d(x,x')$$

$$\leq \|\ell\|_\infty LK\, d(x,x') + \|\ell\|_\infty L'K\, d(x,x').$$

For the middle term, we construct a classifier $h'$ from $\bar{h}_q$ such that $\mathrm{Lip}(h') \leq L$ and $V_p(h') \leq \alpha$ using Fact 2.2, whereby, because of the optimality of $\bar{h}_p$ on $p$,

$$\begin{aligned}
R_p(\bar{h}_p) - R_p(\bar{h}_q) &= (R_p(\bar{h}_p) - R_p(h')) + (R_p(h') - R_p(\bar{h}_q)) \\
&\leq R_p(h') - R_p(\bar{h}_q).
\end{aligned} \tag{8}$$

The construction is as follows: let $\beta \in [0,1]$ to be determined, and

$$h'(x) = \beta(1,0,\ldots,0) + (1-\beta)\bar{h}_q(x),$$

in other words, $h'$ interpolates between the constant classifier that always outputs 0, and the original $\bar{h}_q$. We verify that it is $L$-Lipschitz: $|h'(x)-h'(x')| = (1-\beta)|\bar{h}_q(x)-\bar{h}_q(x')| \leq (1-\beta)L\, d(x,x')$. For its fairness violation on $p$, by Eq. (4),

$$\begin{aligned}
V_p(h') &= \max_{\substack{a,a'\in\mathcal{A}\\c\in\{1,\ldots,C\}}} \left| \int_{\mathcal{X}} h'(x)_k\big(p_{X|a,z_c}(x) - p_{X|a',z_c}(x)\big)\,\mathrm{d}x \right| \\
&\leq (1-\beta) \max_{\substack{a,a'\in\mathcal{A}\\c\in\{1,\ldots,C\}}} \left| \int_{\mathcal{X}} \bar{h}_q(x)\big(p_{X|a,z_c}(x) - p_{X|a',z_c}(x)\big)\,\mathrm{d}x \right| \\
&= (1-\beta)V_p(\bar{h}_q) \\
&\leq (1-\beta)(\alpha+\varepsilon)
\end{aligned}$$

by the assumption that $V_p(\bar{h}_q) \leq \alpha + \varepsilon$, and we have $(1-\beta)(\alpha+\varepsilon) \leq \alpha$ via setting

$$\beta = \frac{\varepsilon}{\alpha+\varepsilon}.$$

Then to bound Eq. (8),

$$\begin{aligned}
&R_p(h') - R_p(\bar{h}_q) \\
&= \int_{\mathcal{X}\times\mathcal{Y}} \sum_{k\in\mathcal{Y}} \ell(y,k)\big(h'(x)_k - \bar{h}_q(x)_k\big)p_{X,Y}(x,y)\,\mathrm{d}xy \\
&= \beta \int_{\mathcal{X}\times\mathcal{Y}} \ell(y,0)p_{X,Y}(x,y)\,\mathrm{d}xy - \beta \int_{\mathcal{X}\times\mathcal{Y}} \sum_{k\in\mathcal{Y}} \ell(y,k)\bar{h}_q(x)_k p_{X,Y}(x,y)\,\mathrm{d}xy \\
&\leq \beta\|\ell\|_\infty.
\end{aligned}$$

The final bound in the statement is obtained by putting the above together. $\qquad\square$

*Proof of Corollary 3.3 Part 1* (Statistical Parity). We pick up from Eq. (8) in the above proof of Theorem 3.2; the next step is to construct a classifier $h'$ such that $\mathrm{Lip}(h') \leq L$ and $V_p(h') \leq \alpha$.

Let $\mu_a \in \Delta(\mathcal{Y})$ denote the class output distribution of $\bar{h}_q$ on the source distribution $p$ conditioned on group $A = a$, that is, $\mu_{a,k} = \mathbb{E}_{X \sim p_X}[\bar{h}_q(X)_k \mid A = a]$, and similarly let $\nu_{a,k} = \mathbb{E}_{X \sim q_X}[\bar{h}_q(X)_k \mid A = a]$ denote that on the target distribution $q$. We will construct an $h'$ off $\bar{h}_q$ such that its conditional output distributions on $p$ is the same as $\nu$ (i.e., that of $\bar{h}_q$ on $q$), which satisfies fairness.

Define
$$d_{a,k} = \max(0, \nu_{a,k} - \mu_{a,k}), \quad s_{a,k} = \frac{\max(0, \mu_{a,k} - \nu_{a,k})}{\mu_{a,k}},$$
then we construct
$$h'(x, a) = \bar{h}_q(x) \odot (1 - s_a) + d_a,$$
where $\odot$ denotes element-wise multiplication. The intuition is to consider the difference between the output distribution of $\bar{h}_q$ on $p$ (which is $\mu$) and the desired target output distribution $\nu$, and construct $h'$ from $\bar{h}_q$ simply by redirecting class assignments going to classes $k$ where $\mu_k > \nu_k$ (i.e., over-target) to classes $j$ where $\mu_j < \nu_j$ (i.e., under-target) uniformly.

We verify that the conditional output distributions of $h'$ on $p$ is indeed $\nu$ (which satisfies fairness): for any $a, k$,
$$\begin{aligned}
\mathbb{E}_{X \sim p_X}&[h'(X, a)_k \mid A = a] \\
&= (1 - s_{a,k}) \mathbb{E}_{X \sim p_X}[\bar{h}_q(X)_k \mid A = a] + d_a \\
&= (1 - s_{a,k}) \mu_{a,k} + d_a \\
&= \mu_{a,k} - \max(0, \mu_{a,k} - \nu_{a,k}) + \max(0, \nu_{a,k} - \mu_{a,k}) \\
&= \nu_{a,k}.
\end{aligned}$$

Moreover, $\mathrm{Lip}(h') \leq L$ because it is derived from $\bar{h}_q$, which is Lipschitz, by multiplying with a number less than 1 and adding a constant.

Then to bound Eq. (8),
$$\begin{aligned}
R_p(h') &- R_p(\bar{h}_q) \\
&= \int_{\mathcal{X} \times \mathcal{Y} \times \mathcal{A}} \sum_{k \in \mathcal{Y}} \ell(y, k) \big(h'(x, a)_k - \bar{h}_q(x)_k\big) p_{X,Y,A}(x, y, a) \, \mathrm{d}xya \\
&= \int_{\mathcal{X} \times \mathcal{Y} \times \mathcal{A}} \sum_{k \in \mathcal{Y}} \ell(y, k) \big(d_{a,k} - s_{a,k} \bar{h}_q(x)_k\big) p_{X,Y,A}(x, y, a) \, \mathrm{d}xya \\
&= \int_{\mathcal{Y} \times \mathcal{A}} \sum_{k \in \mathcal{Y}} \ell(y, k) (d_{a,k} - s_{a,k} \mu_{a,k}) p_{Y,A}(y, a) \, \mathrm{d}ya \\
&= \int_{\mathcal{Y} \times \mathcal{A}} \sum_{k \in \mathcal{Y}} \ell(y, k) (\nu_{a,k} - \mu_{a,k}) p_{Y,A}(y, a) \, \mathrm{d}ya \\
&\leq \varepsilon K \|\ell\|_\infty,
\end{aligned}$$
where the last line follows from Hölder's inequality and $|\nu_{a,k} - \mu_{a,k}| \leq \varepsilon$, because by Lemma B.1, $\varepsilon$ upper bounds the change in group fairness statistics under distribution shift. The remainder of the proof follows from the rest of the proof of Theorem 3.2. $\qquad\square$

To prove the second result of Corollary 3.3 for binary-class EO, we first recall two facts regarding the true positive rate (TPR) and false positive rate (FPR) of randomized binary classifiers. The first fact simply says that both TPR and FPR of the classifier that always output class 1 with probability $\beta$ equal to $\beta$. This means all points on the main diagonal of the ROC plot are achievable by some randomized classifier.

**Fact C.1.** *Let $\beta \in [0, 1]$, then the classifier $h$ such that $h_0(x) = 1 - \beta$, $h_1(x) = \beta$ for all $x \in \mathcal{X}$ has the same TPR and FPR of $\beta$:*
$$\mathrm{TPR}(h) = \mathbb{E}[h_1(X) \mid Y = 1] = \int_{\mathcal{X}} \beta \, \mathbb{P}(X = x \mid Y = 1) \, \mathrm{d}x = \beta,$$
$$\mathrm{FPR}(h) = \mathbb{E}[h_1(X) \mid Y = 0] = \int_{\mathcal{X}} \beta \, \mathbb{P}(X = x \mid Y = 0) \, \mathrm{d}x = \beta.$$

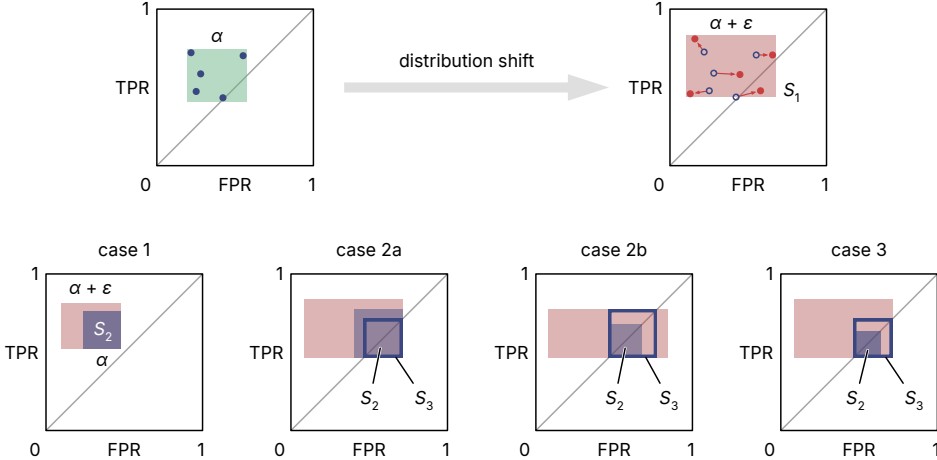

Figure 4: Picture for the cases considered in the proof of Corollary 3.3 Part 2.

The second fact states the linearity of TPR and FPR in $h$:

**Fact C.2.** *Let $h_1, h_2$ be two classifiers, and $\lambda \in [0, 1]$. Let $\mu^{\mathrm{TPR}}, \mu^{\mathrm{FPR}}$ denote the TPR and FPR of $h_1$, respectively, and $\nu^{\mathrm{TPR}}, \nu^{\mathrm{FPR}}$ for those of $h_2$. Then the TPR and FPR of $h = \lambda h_1 + (1 - \lambda)h_2$ are $\lambda\mu^{\mathrm{TPR}} + (1 - \lambda)\nu^{\mathrm{TPR}}$ and $\lambda\mu^{\mathrm{FPR}} + (1 - \lambda)\nu^{\mathrm{FPR}}$.*

*Proof of Corollary 3.3 Part 2* (Equalized Odds, Binary Classification). Let $\mu_a^{\mathrm{TPR}}$ denote the TPR of $\bar{h}_q$ on the source distribution $p$ conditioned on group $A = a$, that is, $\mu_a^{\mathrm{TPR}} = \mathbb{E}_{X \sim p_X}[\bar{h}_q(X)_1 \mid A = a, Y = 1]$, and $\mu_a^{\mathrm{FPR}} = \mathbb{E}_{X \sim p_X}[\bar{h}_q(X)_1 \mid A = a, Y = 0]$ for the conditional FPRs. Let $\overline{\mu}^{\mathrm{TPR}} = \max_a \mu_a^{\mathrm{TPR}}$ denote the maximum conditional TPR, $\underline{\mu}^{\mathrm{TPR}} = \min_a \mu_a^{\mathrm{TPR}}$ the minimum TPR, and analogously define $\overline{\mu}^{\mathrm{FPR}}, \underline{\mu}^{\mathrm{FPR}}$.

We will consider the ROC plot (which plots the FPR on the horizontal axis and TPR on the vertical axis), since the goal of EO fairness is to constrain the group-conditional TPRs and FPRs within a square of side length at most $\alpha$ (Hardt et al. (2016) also based their analysis on the ROC plot).

Define the rectangle $S_1$ on the ROC plot with vertices at:

$$S_1^{\mathrm{UL}} = (\underline{\mu}^{\mathrm{FPR}}, \overline{\mu}^{\mathrm{TPR}}), \qquad\qquad S_1^{\mathrm{UR}} = (\overline{\mu}^{\mathrm{FPR}}, \overline{\mu}^{\mathrm{TPR}}),$$
$$S_1^{\mathrm{BL}} = (\underline{\mu}^{\mathrm{FPR}}, \underline{\mu}^{\mathrm{TPR}}), \qquad\qquad S_1^{\mathrm{BR}} = (\overline{\mu}^{\mathrm{FPR}}, \underline{\mu}^{\mathrm{TPR}}).$$

This rectangle contains the group-conditional TPRs and FPRs of $\bar{h}_q$ on $p$; by the assumption that $V_p(\bar{h}_q) \le \alpha + \varepsilon$, the side lengths of this rectangle are no more than $\alpha + \varepsilon$.

Next, we define a square $S_2$ with side length $\alpha$ contained in $S_1$; later, we will construct $h'$ such that its group-conditional TPRs and FPRs are contained in $S_2$. We consider three cases (three other symmetric cases are omitted); see Fig. 4 for a picture:

1. If $S_1$ is located above and does not intersect with the diagonal line $\{(t, t) : t \in \mathbb{R}\}$, then let the vertices of $S_2$ be

$$S_2^{\mathrm{UL}} = S_1^{\mathrm{BR}} + (-\alpha, \alpha), \qquad\qquad S_2^{\mathrm{UR}} = S_1^{\mathrm{BR}} + (0, \alpha),$$
$$S_2^{\mathrm{BL}} = S_1^{\mathrm{BR}} + (-\alpha, 0), \qquad\qquad S_2^{\mathrm{BR}} = S_1^{\mathrm{BR}}.$$

2. If $S_2$ intersects the diagonal line on the BL-BR side at $(s, s)$, and either the UR-BR or UL-UR side at $(t, t)$, then we construct another square $S_3$ (which is contained in $S_1$) with the following vertices, then consider two cases;

$$S_3^{\mathrm{UL}} = (s, t), \qquad\qquad S_3^{\mathrm{UR}} = (t, t),$$
$$S_3^{\mathrm{BL}} = (s, s), \qquad\qquad S_3^{\mathrm{BR}} = (t, s).$$

If the side length of $S_3$ is less than or equal $\alpha$, then let $S_2$ be the only eligible square in $S_1$ that contains $S_3$:

$$S_2^{\text{UL}} = S_3^{\text{BR}} + (-\alpha, \alpha), \qquad\qquad S_2^{\text{UR}} = S_3^{\text{BR}} + (0, \alpha),$$
$$S_2^{\text{BL}} = S_3^{\text{BR}} + (-\alpha, 0), \qquad\qquad S_2^{\text{BR}} = S_3^{\text{BR}}.$$

3. If $S_2$ intersects the diagonal line as above, but the side length of $S_3$ is greater than $\alpha$, then let the vertices of $S_2$ be

$$S_2^{\text{UL}} = S_3^{\text{BL}} + (0, \alpha), \qquad\qquad S_2^{\text{UR}} = S_3^{\text{BL}} + (\alpha, \alpha),$$
$$S_2^{\text{BL}} = S_3^{\text{BL}}, \qquad\qquad S_2^{\text{BR}} = S_3^{\text{BL}} + (\alpha, 0).$$

It is clear that for any point $u \in S_1 \setminus S_2$, the line that passes through it and its projection $\Pi_{S_2}(u)$ on $S_2$ will intersect the diagonal segment $\{(t,t) : t \in [0,1]\}$, and the $\ell_\infty$ distance between $u$ and $\Pi_{S_2}(u)$ is no more than $\varepsilon$.

Then we construct $h'$ as follows. The strategy is to modify each group-wise component of $\bar{h}_q$ such that the conditional $(\text{FPR}, \text{TPR})$ pair after the modification are in $S_2$. If $\mu_a \in S_2$ already, we let $h'(x, a) = \bar{h}_q(x)$. Otherwise, let $(t, t)$ be the point on the diagonal that intersects with the line that passes through points $\mu_a$ and $\Pi_{S_2}(\mu_a)$, and we know from Facts C.1 and C.2 that there exist $\lambda_a$ and $h_a$ (whose TPR and FPR are on the diagonal) such that the conditional FPR and TPR of $h_a \lambda_a + (1 - \lambda_a)\bar{h}_q$ on $p$ is $\Pi_{S_2}(\mu_a)$, which is what we will set $h'(\cdot, a)$ to. Clearly, $h'$ maintains the Lipschitz property.

Then to bound Eq. (8), we use the fact that the risk of a classifier can be expressed in terms of its (conditional) TPR and FPR:

$$R_p(h) = \sum_{a \in \mathcal{A}} p(A = a)\big(\ell(0,0)(1 - \text{FPR}_a(h)) + \ell(0,1)\text{FPR}_a(h) \\ + \ell(1,0)(1 - \text{TPR}_a(h)) + \ell(1,1)\text{TPR}_a(h)\big),$$

then because the conditional TPRs and FPRs of $h'$ on $p$ is within $\varepsilon$ distance of those of $\bar{h}_q$,

$$R_p(h') - R_p(\bar{h}_q) \leq \varepsilon \sum_{a \in \mathcal{A}} p(A = a)(\ell(0,0) + \ell(0,1) + \ell(1,0) + \ell(1,1)) \leq 4\varepsilon\|\ell\|_\infty. \qquad \square$$

*Proof of Example 1.* We first verify the distribution shift:

$$D_{\text{TV}}(p_{X|A=0}, q_{X|A=0}) = \frac{1}{2} \sum_{x=0}^{1} |p(X = x \mid A = 0) - q(X = x \mid A = 0)|$$
$$= \left|\frac{1 - \alpha - \varepsilon}{2} - \frac{1 - \alpha}{2}\right| = \frac{\varepsilon}{2},$$

and similarly for $D_{\text{TV}}(p_{X|A=1}, q_{X|A=1})$.

Next, the Bayes-optimal fair classifier $\bar{h}_q$ on $q$ coincides with the Bayes-optimal classifier, which is the function $\bar{h}_q(x)_k = \mathbb{1}[x = k]$: it always outputs class 0 on $x = 0$, and class 1 on $x = 1$, and has an error rate of 0. We verify that it satisfies $\alpha$-approximate statistical parity: by Eq. (4),

$$V_q^{\text{SP}}(\bar{h}_q) = \left|\sum_{x=0}^{1} \bar{h}_q(x)_0(q(X = x \mid A = 0) - q(X = x \mid A = 1))\right|$$
$$= |q(X = 0 \mid A = 0) - q(X = 0 \mid A = 1)|$$
$$= \left|\frac{1 - \alpha}{2} - \frac{1 + \alpha}{2}\right| = \alpha.$$

For the Bayes-optimal fair classifier $\bar{h}_p$ on $p$, we derive its error rate as follows. Denote its conditional probability of outputting class 1 on input 0 by $\pi_0 = \bar{h}_p(0)_1$, and that on input 1 by $\pi_1 = \bar{h}_p(1)_1$. We can express its error rate as

$$R(\bar{h}_p) = \frac{1}{2}\pi_0 + \frac{1}{2}(1 - \pi_1) = \frac{1}{2} + \frac{1}{2}(\pi_0 - \pi_1),$$

and its statistical parity violation as

$$V_p^{\text{SP}}(\bar{h}_p) = \left| \sum_{x=0}^{1} \bar{h}_p(x)_1 (p(X = x \mid A = 0) - p(X = x \mid A = 1)) \right|$$

$$= |\pi_0 (p(X = 0 \mid A = 0) - p(X = 0 \mid A = 1))$$
$$+ \pi_1 (p(X = 1 \mid A = 0) - p(X = 1 \mid A = 1))|$$
$$=: |\pi_0 (p_{00} - p_{01}) + \pi_1 (p_{10} - p_{11})|$$
$$= |\pi_0 (p_{00} - p_{01}) + \pi_1 (1 - p_{00} - (1 - p_{01}))|$$
$$= |\pi_0 (p_{00} - p_{01}) - \pi_1 (p_{00} - p_{01})|$$
$$= |(\pi_0 - \pi_1)(p_{00} - p_{01})|$$
$$= |\pi_0 - \pi_1| p_{00} - p_{01}$$
$$= (\alpha + \varepsilon)|\pi_0 - \pi_1|.$$

Then the error rate of $\bar{h}_p$ is the solution to the problem

$$\min_{\pi_0, \pi_1 \in [0,1]} \frac{1}{2} + \frac{1}{2}(\pi_0 - \pi_1) \quad \text{s.t.} \quad |\pi_0 - \pi_1| \leq \frac{\alpha}{\alpha + \varepsilon};$$

it is immediate that an optimal solution is $\pi_0 = 0$ and $\pi_1 = \alpha/(\alpha + \varepsilon)$, so the error rate is $\varepsilon/2(\alpha + \varepsilon)$, which is also the excess risk as 0 is the error rate of $\bar{h}_q$. $\qquad \square$

## D  PROOFS FOR LINEARPOST

First, we review and provide the derivation for the post-processing weights of the single-distribution LinearPost (Theorem 4.1) in terms of the optimal dual values of a linear program (LP). We then extend this derivation for the multiple distribution setting and prove Theorem 4.2.

### D.1  SINGLE-DISTRIBUTION LINEARPOST

Recall the single-distribution fair classification problem,

$$\arg \min_{h} R_p(h) \quad \text{s.t.} \quad V_p(h) \leq \alpha.$$

By Eqs. (2) and (4), it can be expressed as the following linear program with variables $h \in \mathbb{R}^{|\mathcal{X}| \times K}$ and $t \in \mathbb{R}^C$:

$$\min_{h \geq 0, t} \int_{\mathcal{X}} \sum_{k \in \mathcal{Y}} r_p(x)_k h(x)_k p_X(x) \, dx$$

$$\text{s.t.} \quad \sum_{k \in \mathcal{Y}} h(x)_k = 1, \qquad\qquad\qquad \forall x \in \mathcal{X}, \tag{9}$$

$$\left| \int_{\mathcal{X}} h(x)_{y_c} \frac{p_{A,Z|x}(a, z_c)}{p_{A,Z}(a, z_c)} p_X(x) \, dx - t_c \right| \leq \frac{\alpha}{2}, \quad \forall a \in \mathcal{A}, c \in [C],$$

where the first constraint ensures that $h$ represents a valid randomized classifier (row-stochasticity), and $t$ are auxiliary variables introduced to reduce the number of constraints (each $t_c$ will be optimized to the midpoint between the two most violating groups).

Introduce dual variables $\phi : \mathcal{X} \to \mathbb{R}$ and $\psi \in \mathbb{R}^{G \times C}$, then the dual problem of the above is

$$\min_{\phi, \psi} \int_{\mathcal{X}} \phi(x) p_X(x) \, dx - \frac{\alpha}{2} \sum_{c \in [C]} \sum_{a \in \mathcal{A}} |\psi_{a,c}|$$

$$\text{s.t.} \quad \sum_{a \in \mathcal{A}} \psi_{a,c} = 0, \qquad\qquad\qquad \forall c \in [C],$$

$$\phi(x) + \sum_{c: y_c = k} \sum_{a \in \mathcal{A}} \psi_{a,c} \frac{p_{A,Z|x}(a, z_c)}{p_{A,Z}(a, z_c)} \leq r_p(x)_k, \quad \forall x \in \mathcal{X}, k \in \mathcal{Y}.$$

Let $\psi^*$ denote the optimal dual variable, Xian & Zhao (2024) show that the weights for the linear post-processing in Theorem 4.1 are given by

$$\beta_{k,a,z} = -\sum_{c \in [C]} \mathbb{1}[k = y_c, z = z_c] \frac{\psi^*_{a,c}}{p_{A,Z}(a, z_c)}.$$

## D.2   MULTIPLE-DISTRIBUTION LINEARPOST

Let $p$ and $q_1, \ldots, q_M$ be distributions. Similar to above, the multiple-distribution fair classification problem,

$$\arg\min_h R_p(h) \quad \text{s.t.} \quad V_p(h) \le \alpha, \; V_{q_1}(h) \le \alpha, \; \ldots, \; V_{q_M}(h) \le \alpha,$$

can be expressed as the following linear program:

$$
\begin{aligned}
\min_{h \ge 0, t} \quad & \int_{\mathcal{X}} \sum_{k \in \mathcal{Y}} r_p(x)_k h(x)_k p_X(x) \, dx \\
\text{s.t.} \quad & \sum_{k \in \mathcal{Y}} h(x)_k = 1, && \forall x \in \mathcal{X}, \\
& \left| \int_{\mathcal{X}} h(x)_{y_c} \frac{p_{A,Z|x}(a, z_c)}{p_{A,Z}(a, z_c)} p_X(x) \, dx - t_{0,c} \right| \le \frac{\alpha}{2}, && \forall a \in \mathcal{A}, c \in [C], \\
& \left| \int_{\mathcal{X}} h(x)_{y_c} \frac{q_{1\,A,Z|x}(a, z_c)}{q_{1\,A,Z}(a, z_c)} q_{1\,X}(x) \, dx - t_{1,c} \right| \le \frac{\alpha}{2}, && \forall a \in \mathcal{A}, c \in [C], \\
& \qquad\qquad\qquad\qquad \vdots \\
& \left| \int_{\mathcal{X}} h(x)_{y_c} \frac{q_{M\,A,Z|x}(a, z_c)}{q_{M\,A,Z}(a, z_c)} q_{M\,X}(x) \, dx - t_{M,c} \right| \le \frac{\alpha}{2}, && \forall a \in \mathcal{A}, c \in [C].
\end{aligned}
\tag{10}
$$

To derive the dual, we introduce dual variables $\phi : \mathcal{X} \to \mathbb{R}$ and $\psi^+, \psi^- \in \mathbb{R}^{(M+1) \times G \times C}$. The Lagrangian is

$$
\begin{aligned}
L(h, &t, \phi, \psi^+, \psi^-) \\
&= \int_{\mathcal{X}} \sum_{k \in \mathcal{Y}} r_p(x)_k h(x)_k p_X(x) \, dx + \int_{\mathcal{X}} \left( 1 - \sum_{k \in \mathcal{Y}} h(x)_k \right) p_X(x) \phi(x) \, dx \\
&\quad + \sum_{c \in [C]} \sum_{a \in \mathcal{A}} \left( -\frac{\alpha}{2} + t_{0,c} - \int_{\mathcal{X}} h(x)_{y_c} \frac{p_{A,Z|x}(a, z_c)}{p_{A,Z}(a, z_c)} p_X(x) \, dx \right) \psi^+_{0,a,c} \\
&\quad + \sum_{c \in [C]} \sum_{a \in \mathcal{A}} \left( -\frac{\alpha}{2} - t_{0,c} + \int_{\mathcal{X}} h(x)_{y_c} \frac{p_{A,Z|x}(a, z_c)}{p_{A,Z}(a, z_c)} p_X(x) \, dx \right) \psi^-_{0,a,c} \\
&\quad + \sum_{m=1}^{M} \sum_{c \in [C]} \sum_{a \in \mathcal{A}} \left( -\frac{\alpha}{2} + t_{m,c} - \int_{\mathcal{X}} h(x)_{y_c} \frac{q_{m\,A,Z|x}(a, z_c)}{q_{m\,A,Z}(a, z_c)} q_{m\,X}(x) \, dx \right) \psi^+_{m,a,c} \\
&\quad + \sum_{m=1}^{M} \sum_{c \in [C]} \sum_{a \in \mathcal{A}} \left( -\frac{\alpha}{2} - t_{m,c} + \int_{\mathcal{X}} h(x)_{y_c} \frac{q_{m\,A,Z|x}(a, z_c)}{q_{m\,A,Z}(a, z_c)} q_{m\,X}(x) \, dx \right) \psi^-_{m,a,c},
\end{aligned}
$$

collecting terms,

$$L(h, t, \phi, \psi^+, \psi^-)$$

$$= \int_{\mathcal{X}} \phi(x) p_X(x) \, dx - \sum_{m=0}^{M} \sum_{c \in [C]} \sum_{a \in \mathcal{A}} \left( t_{m,c}(\psi^+_{m,a,c} - \psi^-_{m,a,c}) - \frac{\alpha}{2}(\psi^+_{m,a,c} + \psi^-_{m,a,c}) \right)$$

$$+ \int_{\mathcal{X}} \sum_{k \in \mathcal{Y}} \left( r_p(x)_k - \left( \underbrace{\phi(x) + \sum_{m=0}^{M} \sum_{c: y_c = k} \sum_{a \in \mathcal{A}} \frac{q_{mA,Z|x}(a, z_c)}{q_{mA,Z}(a, z_c)} \frac{q_{mX}(x)}{p_X(x)} (\psi^+_{m,a,c} - \psi^-_{m,a,c}) \right)}_{(\star)} \right) h(x)_k p_X(x) \, dx,$$

where we defined $q_0 = p$.

By strong duality, $\min_{h \geq 0, t} \max_{\phi, \psi^+ \geq 0, \psi^- \geq 0} L = \max_{\phi, \psi^+ \geq 0, \psi^- \geq 0} \min_{h \geq 0, t} L$. Note that, if $r_p(x)_k < (\star)$ for some $(x, k)$, then we can send $L$ to $-\infty$ by setting $\overline{h}(x)_k = \infty$, so we must have that $r_p(x)_k \geq (\star)$ for all $x, k$. But with this constraint, the best we can do for $\min_{h \geq 0, t} L$ is to set $h = 0$, so the last line is omitted. Similarly, we must have $\sum_{a \in \mathcal{A}}(\psi^+_{m,a,c} - \psi^-_{m,a,c}) = 0$ from its interaction with $t_{m,c}$ for all $m, c$.

So the dual problem is

$$\min_{\phi, \psi} \int_{\mathcal{X}} \phi(x) p_X(x) \, dx - \frac{\alpha}{2} \sum_{m=0}^{M} \sum_{c \in [C]} \sum_{a \in \mathcal{A}} |\psi_{m,a,c}|$$

$$\text{s.t.} \quad \sum_{a \in \mathcal{A}} \psi_{m,a,c} = 0, \qquad\qquad\qquad \forall c \in [C], \, m \in [M]$$

$$\phi(x) + \sum_{m=0}^{M} \sum_{c: y_c = k} \sum_{a \in \mathcal{A}} \psi_{m,a,c} \frac{q_{mA,Z|x}(a, z_c)}{q_{mA,Z}(a, z_c)} \frac{q_{mX}(x)}{p_X(x)} \leq r_p(x)_k, \quad \forall x \in \mathcal{X}, k \in \mathcal{Y}.$$

Now, we follow a similar analysis in (Xian & Zhao, 2024) to prove Theorem 4.2.

*Proof of Theorem 4.2.* Let $h^*$ be a minimizer of the primal LP (Eq. (10)) and $\psi^*$ an optimal dual variable. By definition, $h^*$ is the Bayes-optimal randomized fair classifier achieving the minimum risk on $p$ and satisfying fairness on $p, q_1, \ldots, q_M$ simultaneously.

Define

$$r_{\text{fair}}(x, k) = r_p(x)_k - \sum_{m=0}^{M} \sum_{c: y_c = k} \sum_{a \in \mathcal{A}} \psi^*_{m,a,c} \frac{q_{mA,Z|x}(a, z_c)}{q_{mA,Z}(a, z_c)} \frac{q_{mX}(x)}{p_X(x)}$$

$$= r_p(x)_k + \sum_{a \in \mathcal{A}, z \in \mathcal{Z}} \sum_{m=0}^{M} \beta_{m,k,a,z} q_{mA,Z|x}(a, z) \frac{q_{mX}(x)}{p_X(x)}$$

with

$$\beta_{m,k,a,z} = - \sum_{c \in [C]} \mathbb{1}[k = y_c, z = z_c] \frac{\psi^*_{m,a,c}}{q_{mA,Z}(a, z_c)};$$

note that $x \mapsto \arg\min_k r_{\text{fair}}(x, k)$ is the classifier proposed in Theorem 4.2.

Then, the second constraint of the dual problem reads $\phi(x) - r_{\text{fair}}(x, k) \leq 0$ for all $x, k$. By complementary slackness (Papadimitriou & Steiglitz, 1998), $\phi(x) - r_{\text{fair}}(x, k) \iff h^*(x)_k > 0$, with the right hand side meaning that the optimal randomized fair classifier has a non-zero probability of outputting class $k$ on input $x$, and it can be shown that

$$h^*(x)_k > 0 \implies k \in \arg\min_{k \in \mathcal{Y}} r_{\text{fair}}(x, k).$$

To show that the function on the right hand side is equivalent to the optimal fair classifier on the left hand side, we need to establish the "$\iff$" relation (almost surely), which is saying that

the $\arg\min$ is unique (almost surely); this is where the continuity condition helps. Note that $x \mapsto \arg\min_k r_{\text{fair}}(x,k)$ is a $K$-class linear classifier with features $(r_p(x), p_{A,Z|x}, q_{1A,Z|x}, \cdots, q_{MA,Z|x})$, and the class prototypes always have a non-zero component in the $r_p$-features, so ties occur (i.e., $\arg\min$ is non-unique) when the features lie on any of the hyperplanes associated with the class prototypes. The continuity condition simply implies that this occurs with probability zero with respect to $x \sim p_X$ or any of $q_{1X}, \ldots, q_{MX}$, so "$\Longleftrightarrow$" holds almost surely on $p, q_1, \ldots, q_M$. Finally, by strong duality, the proposed function achieves the same risk as $h^*$ and satisfies the same fairness constraints, hence is an optimal fair classifier. $\qquad\square$

## E  EXPERIMENT DETAILS

**Dataset.**   Our experiments are performed on the ACSIncome dataset (Ding et al., 2021), which is based on the UCI Adult dataset (Kohavi, 1996), a standard benchmark in the algorithmic fairness literature. We use data from the 2018 survey year (1-year horizon), partitioned into 51 subsets according to the individual's home U.S. state or territory, and retain the 27 largest subsets by sample size: the largest is California (CA) with 78281 examples, and the smallest among them is Louisiana (LA) with 8240 examples; Florida (FL) has 39541 examples. We apply standard pre-processing for tabular data: categorical features are one-hot encoded, and all features are standardized.

The uncertainty estimates in Figs. 1, 5, 6, 9 and 10 and Table 1 are obtained by averaging over 5 runs with different random seeds for splitting the dataset.

**Reductions.**   The Reductions fair classification algorithm, proposed by Agarwal et al. (2018), is based on a two-player game formulation of the fair classification problem. The algorithm relies on a cost-sensitive classification oracle and uses no-regret learning, and outputs a randomized ensemble of classifiers.

We use the implementation provided in the AIF360 library with default hyperparameters (Bellamy et al., 2018), and sweep the tolerance parameter for the "allowed fairness constraint violation" (eps) from $\{100, 50, 20, 10, 5, 2, 1, 0.5, 0.2, 0.1, 0.05, 0.02, 0.01, 0.005, 0.002, 0.001\}$. The base prediction model is a gradient-boosted decision tree (GBDT), trained using LightGBM with default hyperparameters (Ke et al., 2017). The data is split 70/30 for training and testing.

**LinearPost.**   A technical description of LinearPost is provided in Section 4.1: we follow the "pre-train then post-process" procedure where we first fit a GBDT predictor for $(A, Y)$ given $X$ (which suffices for the fairness criteria we consider, SP, EOpp, and EO, and for using 0-1 loss as the objective), then apply LinearPost to enforce fairness. The data is split 60/10/30 for pre-training, post-processing, and testing. LinearPost involves solving a linear program (LP) with $(NK+C)$ variables and $(N + GC)$ constraints (Definition 2.1), where $N$ is the number of post-processing examples. We use the Gurobi optimizer to solve these LPs.

We sweep the fairness tolerance parameter $\alpha$ over 15 evenly spaced values between $\alpha_{\min} = 0.001$ and $\alpha_{\max}$, where $\alpha_{\max}$ is set to the fairness violation of the unmitigated GBDT base classifier on the test set (of the training/source distribution).

**Robust LinearPost.**   A technical description of robust LinearPost is provided in Section 4, which iteratively alternates between finding a perturbation within the uncertainty set and enforcing fairness with respect to all previously found perturbations using multiple-distribution LinearPost (Section 4.1). Here, we sweep the fairness tolerance parameter $\alpha$ logarithmically between $\alpha_{\min} = 0.001$ and $\alpha_{\max}$.

The tolerance parameter for the pessimization step is set to $\tau = 0.001$, and we limit the number of iterations to $T = 20$ (Algorithm 1). The uncertainty set is implemented using the covariate and concept shift perturbation models described in Section 4.1, parameterized by one-hidden-layer neural nets of width 128 with LeakyReLU activation. The input to these neural nets is the output of the GBDT base model (i.e., the probabilistic predictions of $(A, Y)$ given $X$).

For the pessimization, we optimize the perturbation models to maximize the fairness violation (of the current classifier) via full-batch gradient ascent using Adam (default hyperparameters, learning rate 0.01) for 1000 epochs. To reduce variance, we warm-start training by minimizing the KL divergence

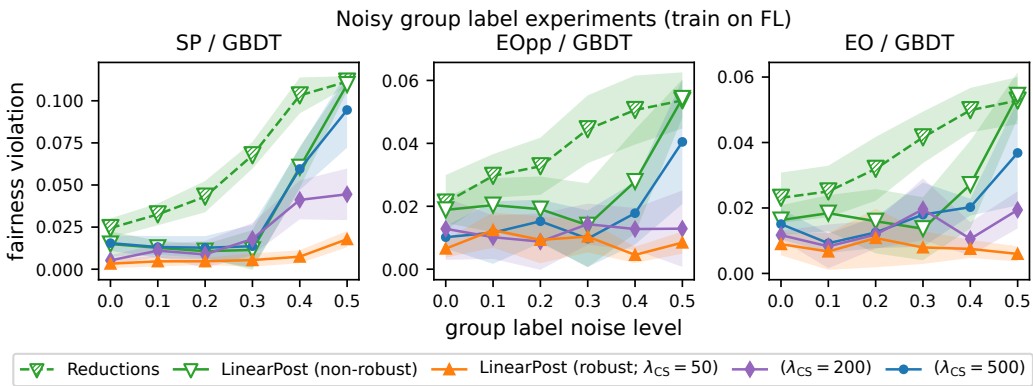

Figure 5: Fairness violations under increasing group label noise level by Reductions, non-robust LinearPost, and robust LinearPost ($\lambda_{\mathrm{CS}} \in \{50, 200, 500\}$, $\lambda_{\mathrm{IW}} = \infty$), each under the tolerance setting that minimizes the average violation. See Table 2 for the selected tolerances.

between $p$ and $q$ for 1000 epochs, and perform 5 trials with different random initializations, selecting the one that induces the highest fairness violation.

For geographic shift experiments (Section 5), we sweep the hyperparameters of the perturbation models over $\lambda_{\mathrm{IW}} \in \{20, 50\}$ and $\lambda_{\mathrm{CS}} \in \{200, 500\}$. Results for the $\lambda_{\mathrm{IW}} = 20$, $\lambda_{\mathrm{IW}} = 500$ configuration are shown in the main text; results for the remaining configurations are in Fig. 11.

## F  EXPERIMENTS FOR NOISY GROUP LABELS

In this set of experiments, we evaluate robust LinearPost under noisy group labels, where the sensitive attribute $A$ is randomly replaced with a uniformly drawn value from $\mathcal{A} = \{0, \ldots, G - 1\}$ on a fraction $\gamma$ of the training data, following Wang et al. (2020). This corresponds to a concept shift between the training (perturbed) and the test (true) distribution, as $p^{\mathrm{perturbed}}(A = a \mid X) = \gamma/G + (1 - \gamma)p^{\mathrm{true}}(A = a \mid X)$. To account for this, we apply the concept shift model from Section 4.2 within our robust LinearPost framework (which in fact generally handles adversarial label noise, not just uniform noise), while disabling the covariate shift model (i.e., $\lambda_{\mathrm{IW}} = \infty$).

We run these experiments on data from Florida (FL) in the ACSIncome dataset, on which the unmitigated GBDT classifier exhibits substantial violations of all three fairness criteria (SP, EOpp, and EO). And, rather than sampling group labels uniformly as described above, we flip the binary sensitive attribute on a random $\gamma$ fraction of the data (referred to as the group label noise level). All other experimental settings are the same as those in Section 5, e.g., the base model is GBDT.

**Results.**  Figure 5 shows the fairness violations under increasing group label noise levels, using each algorithm's best tolerance setting chosen to minimize the macro-average violation on the validation set across noise levels (not necessarily the strictest setting tested). As expected, fairness violations increase with noise level. Robust LinearPost consistently achieves the lowest violations, with weaker regularization settings ($\lambda_{\mathrm{CS}}$) providing better robustness by inducing a larger uncertainty set. These results confirm that the concept shift component of our uncertainty set construction indeed captures such shifts, and that robust LinearPost can, in turn, effectively mitigate their impact.

We plot the accuracy-fairness tradeoffs achieved by robust LinearPost under varying fairness tolerances in Fig. 7, compared to the interpolation between non-robust LinearPost and the constant 0 classifier. Robust LinearPost can achieve tradeoffs that lie above this interpolation line, indicating that its fairness improvements are non-trivial. However, for EO under large noise levels, the tradeoffs are no better than interpolation; we will discuss possible improvements in Appendix G.

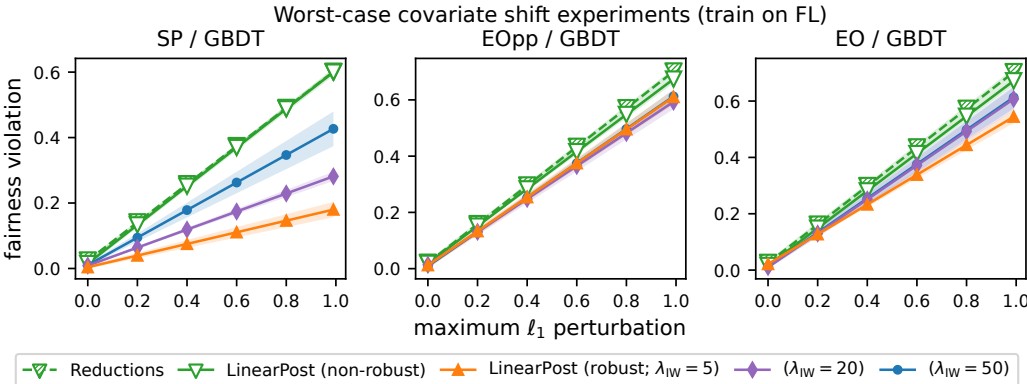

Figure 6: Fairness violations under increasing worst-case bounded covariate shift by Reductions, non-robust LinearPost, and robust LinearPost ($\lambda_{\text{IW}} \in \{5, 20, 50\}$, $\lambda_{\text{CS}} = \infty$), each under the tolerance setting that minimizes the maximum violation. See Table 2 for the selected tolerances.

## G  EXPERIMENTS FOR COVARIATE SHIFT

We evaluate robust LinearPost under covariate shift, where the test distribution's marginal distribution $p_X^{\text{perturbed}}$ differs from the original (true) distribution $p_X^{\text{true}}$ (Mandal et al., 2020).

Classifiers are trained on Florida (FL) data from the ACSIncome dataset using robust LinearPost with the covariate shift model from Section 4.2; the concept shift model is disabled ($\lambda_{\text{CS}} = \infty$). At test time, we evaluate the classifiers under worst-case perturbations to the sample weights of the test examples that maximize the fairness violation, within a bounded $\ell_1$ distance from the uniform empirical distribution ($1/N$). These worst-case perturbations are computed using the code by Mandal et al. (2020).[6] All other experimental settings are the same as those in Section 5.

**Results.**  Figure 6 shows the fairness violations of the evaluated classifiers under increasing magnitudes of adversarial covariate shift, using each algorithm's best tolerance setting chosen to minimize the maximum violation on the validation set (not necessarily the strictest setting tested). Again, as expected, fairness violations grow with perturbation magnitude, and robust LinearPost consistently yields the lowest violations. These results validate the role of the covariate shift component in our uncertainty set construction.

We note that the gains from robust LinearPost are smaller under EOpp and EO fairness than under SP. This may be partly due to the difficulty of the task: on the related Adult dataset, the robust fair algorithm of Mandal et al. (2020) also achieved only modest improvements against worst-case covariate shift for EO fairness. One potential direction for improvement is to strengthen the covariate shift model; for example, by allowing the neural network to take the original features in $\mathcal{X}$ as input, rather than the GBDT outputs used in our current setup (Section 4.2).

In Fig. 8, we plot the accuracy-fairness tradeoffs of robust LinearPost along with the baseline formed by interpolating between the fairest non-robust LinearPost classifier and the constant-0 classifier. Robust LinearPost achieves tradeoffs lying above this baseline, indicating that its fairness improvements are non-trivial. For EOpp fairness, all tested regularization strengths $\lambda_{\text{IW}} \in \{5, 20, 50\}$ yield similar fairness, with weaker regularization lowering accuracy without improving fairness—in particular, under EO, setting $\lambda_{\text{IW}} = 5$ resulted in tradeoffs below the interpolation baseline.

---

[6]https://github.com/samuel-deng/Ensuring-Fairness-Beyond-the-Training-D
ata/tree/f8f59390e78696aaad66a8a5ca4087613fe0255c/main/Part3_Comparisons

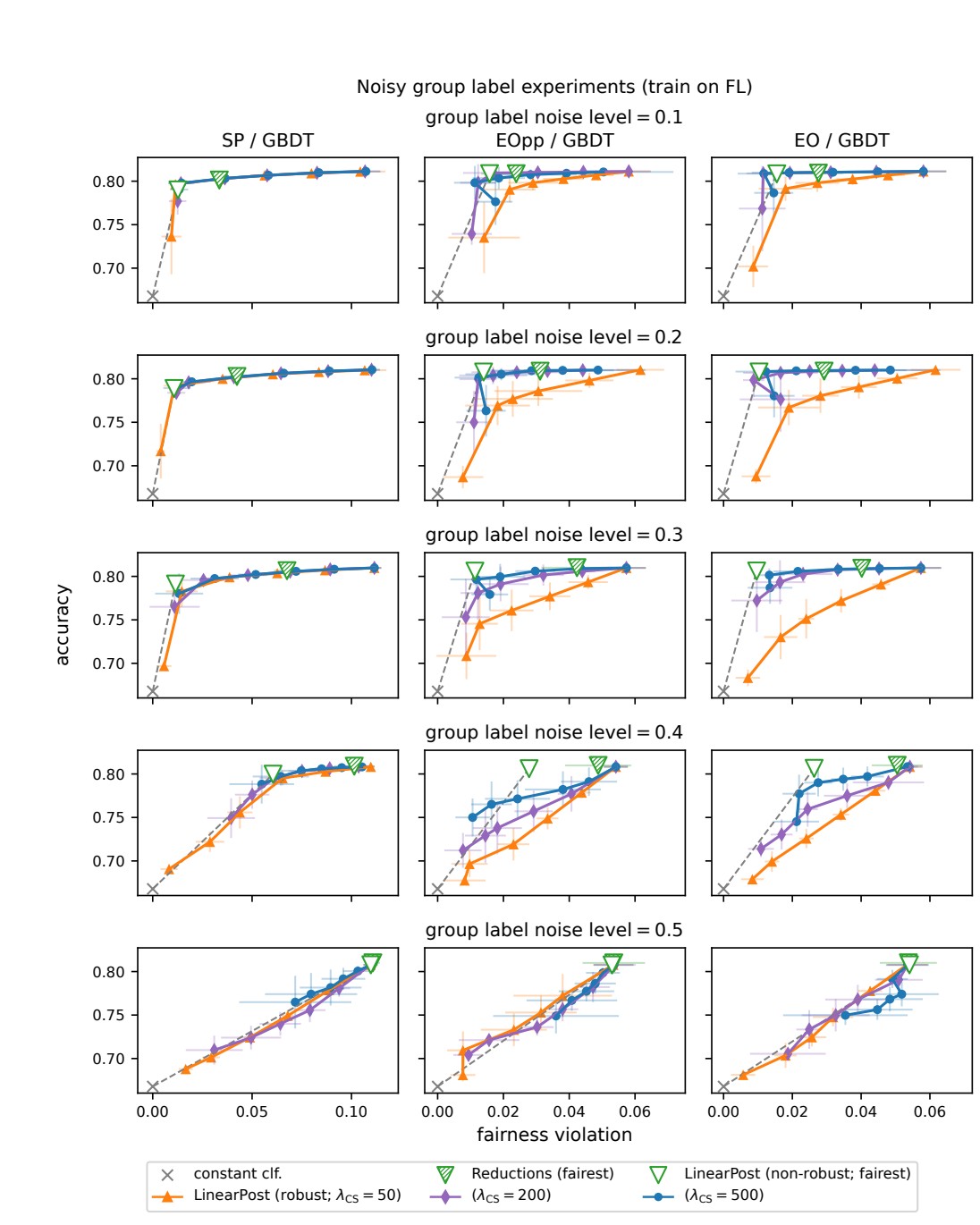

Figure 7: Accuracy-fairness tradeoffs on FL by robust LinearPost trained on FL data with noisy group labels ($\lambda_{\mathrm{CS}} \in \{50, 200, 500\}$, $\lambda_{\mathrm{IW}} = \infty$). For comparison, we include the fairest Pareto-optimal classifiers from non-robust LinearPost and Reductions, as well as the randomized interpolation between the fairer baseline and the constant $0$ classifier (dashed lines).

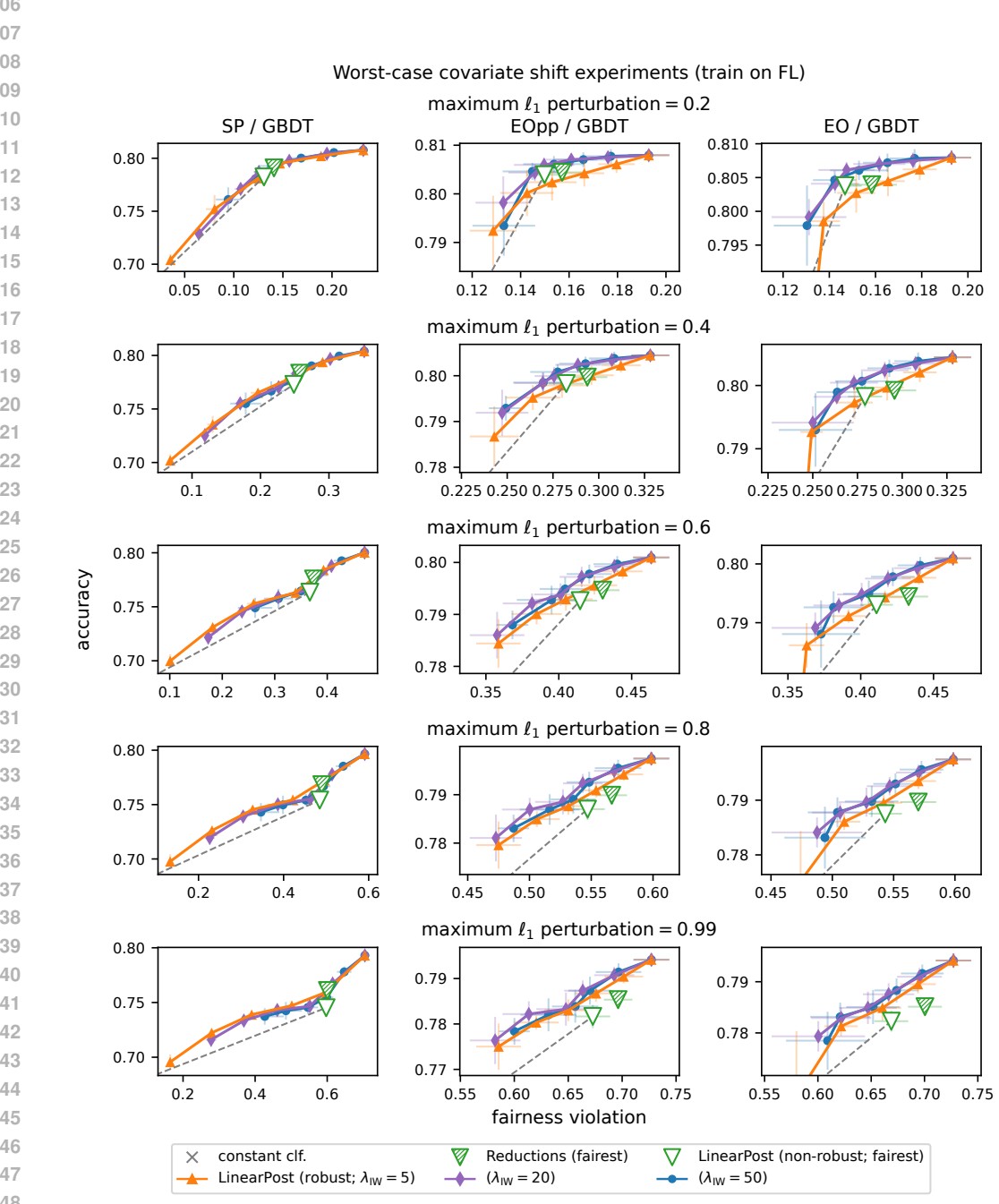

Figure 8: Fairness violations under worst-case bounded covariate shifts by robust LinearPost trained on FL ($\lambda_{\text{IW}} \in \{5, 20, 50\}$, $\lambda_{\text{CS}} = \infty$), showing alongside the accuracies without perturbation. For comparison, we include the fairest Pareto-optimal classifiers from non-robust LinearPost and Reductions, as well as the randomized interpolation between the fairer baseline and the constant $0$ classifier (dashed lines).

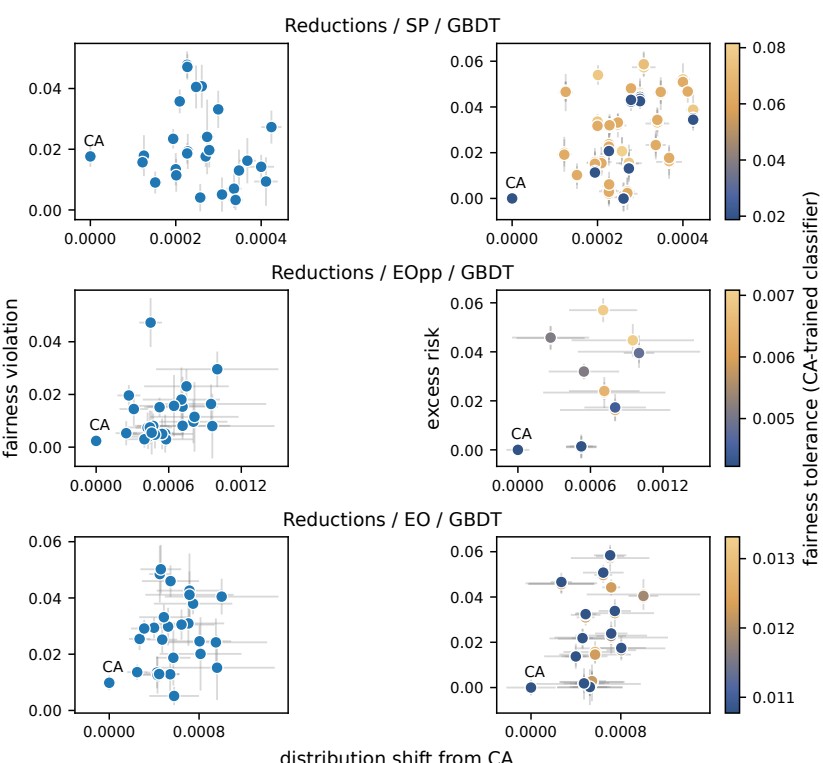

Figure 9: Fairness violation and excess risk on each region by Reductions trained on CA data with GBDT as the base model, under varying tolerances. See the caption of Fig. 1.

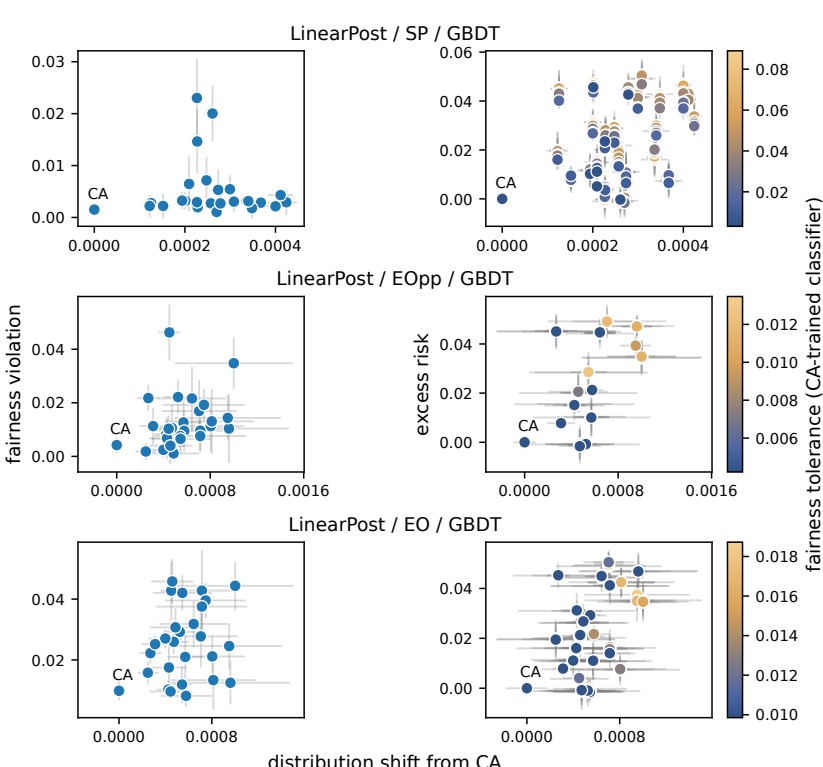

Figure 10: Fairness violation and excess risk on each region by LinearPost trained on CA data with GBDT as the base model, under varying tolerances. See the caption of Fig. 1.

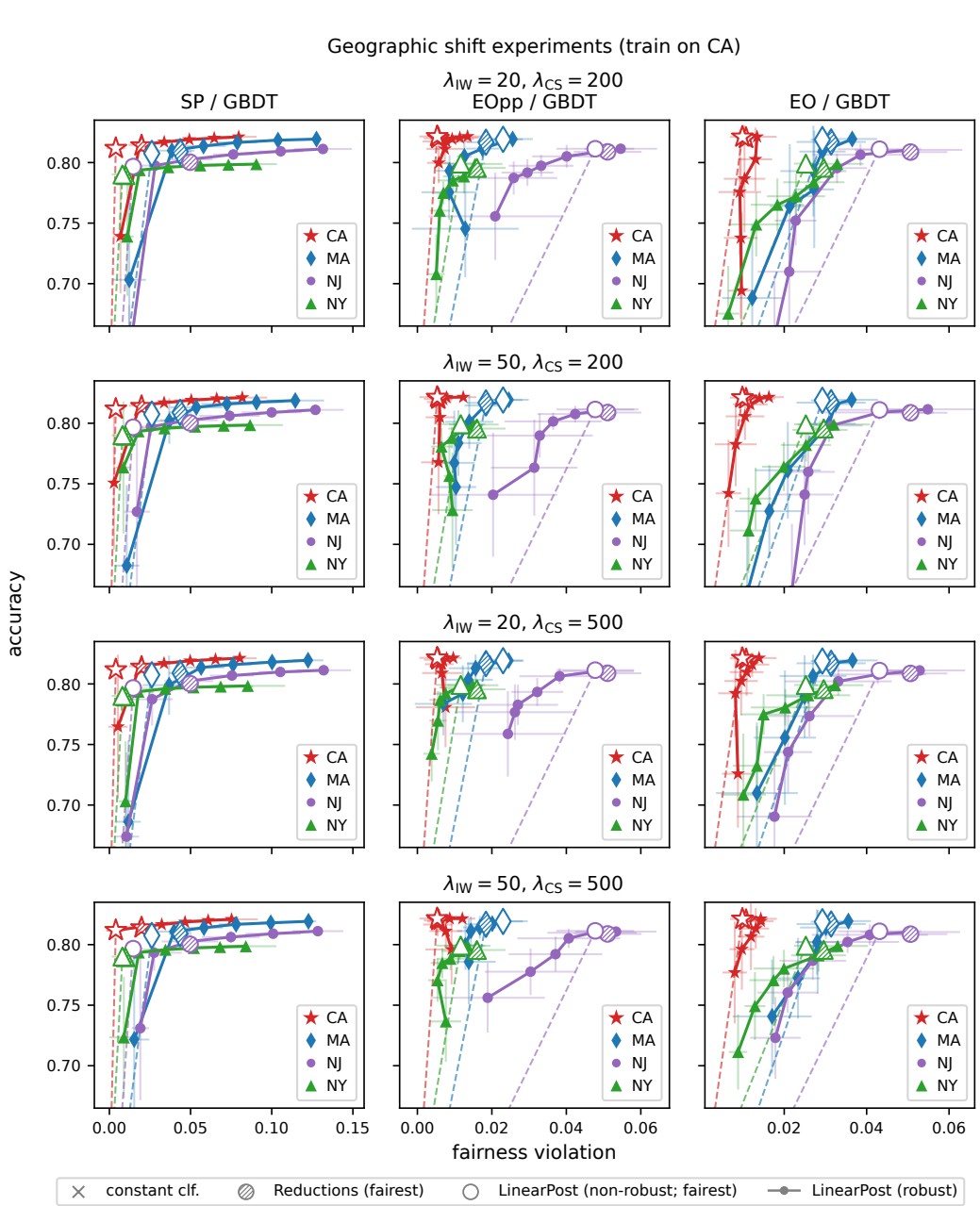

Figure 11: Accuracy-fairness tradeoffs on each region by robust LinearPost trained on CA data (under various $\lambda_{IW}$ and $\lambda_{CS}$ settings). For comparison, we include the fairest Pareto-optimal classifiers from non-robust LinearPost and Reductions, as well as the randomized interpolation between the fairer baseline and the constant 0 classifier (dashed lines).

Table 1: Macro average accuracies and fairnesses by Reductions and LinearPost (non-robust and robust) trained on CA data, under the tolerance setting that minimizes macro average violation. See Fig. 2 for results on all regions. For LinearPost, the tolerance is reported as a percentage of $\alpha$ within $[0.001, \alpha_{\max}]$, where $\alpha_{\max}$ is the violation on CA without post-processing.

| | Geographic Shift Experiments | | |
|---|---|---|---|
| Algorithm | Accuracy (avg.) | Fairness Vio. (avg.) | Selected Tol. |
| *Statistical parity* | | | |
| Reductions | $0.7740 \pm 0.0006$ | $0.0240 \pm 0.0012$ | 2 |
| LinearPost (non-robust) | $0.7721 \pm 0.0006$ | $0.0145 \pm 0.0013$ | 0.0667 |
| (robust; $\lambda_{IW} = 20, \lambda_{CS} = 200$) | $0.6715 \pm 0.0015$ | $0.0089 \pm 0.0014$ | 0.0667 |
| (robust; $\lambda_{IW} = 50, \lambda_{CS} = 200$) | $0.6748 \pm 0.0031$ | $0.0104 \pm 0.0013$ | 0 |
| (robust; $\lambda_{IW} = 20, \lambda_{CS} = 500$) | $0.6777 \pm 0.0017$ | $0.0085 \pm 0.0012$ | 0 |
| (robust; $\lambda_{IW} = 50, \lambda_{CS} = 500$) | $0.7068 \pm 0.0038$ | $0.0130 \pm 0.0020$ | 0 |
| Constant 0 classifier | $0.6315 \pm 0.0006$ | 0 | - |
| *Equal opportunity* | | | |
| Reductions | $0.7819 \pm 0.0006$ | $0.0163 \pm 0.0016$ | 5 |
| LinearPost (non-robust) | $0.7856 \pm 0.0006$ | $0.0160 \pm 0.0015$ | 0 |
| (robust; $\lambda_{IW} = 20, \lambda_{CS} = 200$) | $0.6994 \pm 0.0035$ | $0.0104 \pm 0.0014$ | 0.5333 |
| (robust; $\lambda_{IW} = 50, \lambda_{CS} = 200$) | $0.7047 \pm 0.0040$ | $0.0114 \pm 0.0013$ | 0.4667 |
| (robust; $\lambda_{IW} = 20, \lambda_{CS} = 500$) | $0.6895 \pm 0.0018$ | $0.0107 \pm 0.0014$ | 0.2 |
| (robust; $\lambda_{IW} = 50, \lambda_{CS} = 500$) | $0.7422 \pm 0.0056$ | $0.0134 \pm 0.0016$ | 0.3333 |
| Constant 0 classifier | $0.6315 \pm 0.0006$ | 0 | - |
| *Equalized odds* | | | |
| Reductions | $0.7819 \pm 0.0006$ | $0.0295 \pm 0.0015$ | 0.2 |
| LinearPost (non-robust) | $0.7853 \pm 0.0006$ | $0.0267 \pm 0.0013$ | 0 |
| (robust; $\lambda_{IW} = 20, \lambda_{CS} = 200$) | $0.6531 \pm 0.0015$ | $0.0168 \pm 0.0016$ | 0 |
| (robust; $\lambda_{IW} = 50, \lambda_{CS} = 200$) | $0.6655 \pm 0.0021$ | $0.0187 \pm 0.0018$ | 0 |
| (robust; $\lambda_{IW} = 20, \lambda_{CS} = 500$) | $0.7465 \pm 0.0036$ | $0.0223 \pm 0.0019$ | 0.4 |
| (robust; $\lambda_{IW} = 50, \lambda_{CS} = 500$) | $0.7347 \pm 0.0045$ | $0.0212 \pm 0.0017$ | 0.2 |
| Constant 0 classifier | $0.6315 \pm 0.0006$ | 0 | - |

Table 2: Tolerance settings of each algorithm for the results in Figs. 5 and 6. For LinearPost, the tolerance is reported as a percentage of $\alpha$ within $[0.001, \alpha_{\max}]$, where $\alpha_{\max}$ is the violation on CA without post-processing.

| Noisy Group Label Experiments | | Worst-Case Covariate Shift Experiments | |
|---|---|---|---|
| Algorithm | Selected Tol. | Algorithm | Selected Tol. |
| *Statistical parity* | | | |
| Reductions | 0.2 | Reductions | 0.5 |
| LinearPost (non-robust) | 0 | LinearPost (non-robust) | 0 |
| LinearPost (robust; $\lambda_{CS} = 50$) | 0 | LinearPost (robust; $\lambda_{IW} = 5$) | 0 |
| LinearPost (robust; $\lambda_{CS} = 200$) | 0 | LinearPost (robust; $\lambda_{IW} = 20$) | 0 |
| LinearPost (robust; $\lambda_{CS} = 500$) | 0 | LinearPost (robust; $\lambda_{IW} = 50$) | 0 |
| *Equal opportunity* | | | |
| Reductions | 0.2 | Reductions | 0.2 |
| LinearPost (non-robust) | 0 | LinearPost (non-robust) | 0 |
| LinearPost (robust; $\lambda_{CS} = 50$) | 0.1333 | LinearPost (robust; $\lambda_{IW} = 5$) | 0.8 |
| LinearPost (robust; $\lambda_{CS} = 200$) | 0.0667 | LinearPost (robust; $\lambda_{IW} = 20$) | 0.4 |
| LinearPost (robust; $\lambda_{CS} = 500$) | 0 | LinearPost (robust; $\lambda_{IW} = 50$) | 0.1333 |
| *Equalized odds* | | | |
| Reductions | 0.002 | Reductions | 10 |
| LinearPost (non-robust) | 0.002 | LinearPost (non-robust) | 0 |
| LinearPost (robust; $\lambda_{CS} = 50$) | 0.0667 | LinearPost (robust; $\lambda_{IW} = 5$) | 0 |
| LinearPost (robust; $\lambda_{CS} = 200$) | 0 | LinearPost (robust; $\lambda_{IW} = 20$) | 0.3333 |
| LinearPost (robust; $\lambda_{CS} = 500$) | 0 | LinearPost (robust; $\lambda_{IW} = 50$) | 0.1333 |

