# OpenReview forum: "Group Fairness Under Distribution Shifts: Analysis and Robust Post-Processing"
_ICLR.cc/2026/Conference — Submitted to ICLR 2026_

### Official Review · Reviewer_CTc5 · 2025-10-29

**Soundness:** 3
**Presentation:** 3
**Contribution:** 2
**Rating:** 2
**Confidence:** 4

**Summary:**

This paper studies how group fairness guarantees are getting worse under distribution shifts and proposes a robust post-processing algorithm to address this issue. The authors first provide a theoretical analysis for randomized fair classifiers, deriving upper bounds on fairness violation and excess risk when the test distribution differs from training. This result also showed that the bound can be decomposed into covariate and concept shifts. Thus, their results reveal that smoother classifiers are more robust to such shifts. Building on this insight, they extend the LinearPost algorithm into a robust variant that iteratively learns classifiers satisfying fairness constraints across multiple perturbed distributions within an uncertainty set modeling potential shifts. Experiments on the ACSIncome dataset under geographic shifts demonstrate that the proposed method significantly improves fairness on unseen regions—albeit at some cost to accuracy—showing that explicit robustness modeling can enhance the transferability of fairness guarantees.

**Strengths:**

* The paper provides a rigorous theoretical analysis of how fairness degrades under domain shifts, deriving upper bounds on fairness violation and excess risk. The result also offers insightful decomposition of the fairness degradation into covariate shift and concept shift, helping clarify which aspects of distribution change most strongly affect fairness guarantees.
* The work extends the LinearPost algorithm to a robust version capable of handling multiple worst-case distributions simultaneously, allowing fairness constraints to generalize across an uncertainty set.
• The proposed method shows strong empirical robustness on the ACSIncome (Adult) dataset, achieving improved fairness across unseen geographic regions compared to existing baselines.

**Weaknesses:**

* It is unclear how much novel theoretical contribution the analysis provides beyond existing results. The first part of Theorem 3.1 reproduces known bounds, and while the second part introduces a new decomposition, the resulting insights are not fully explored or clearly leveraged in the paper. Similarly, in Theorem 3.2, the first two terms of the excess-risk bound have appeared in prior analyses of domain shifts, and the new term $\varepsilon / (\alpha + \varepsilon)$ seems to have limited influence on robustness, as all three terms are dominant for the upper bound in the inequality. That is, even if the new term is increased or decreased depending on the $\alpha$, the robustness for the risk is less likely to be affected if the last two terms have large values.

* The proposed robust post-processing algorithm appears somewhat disconnected from the theoretical results. Its effectiveness heavily depends on accurately estimating the conditional probabilities of $(A, Z)$ given $X$, which may not be realistic in many domains—especially for high-dimensional or unstructured data such as text or images (e.g., CelebA or CivilComments), where predicting sensitive attributes or fairness-relevant variables is inherently unreliable.

* The experimental evaluation is limited to a single benchmark (ACSIncome), making it difficult to assess the generality of the approach. Additional experiments on diverse datasets or modalities would strengthen the empirical evidence, particularly to test the issues mentioned in the second weakness, the algorithm’s dependence on the choice of uncertainty set Q and its robustness under different types of distribution shifts.

**Questions:**

The robust post-processing algorithm introduces two regularization parameters, $\lambda_{\text{IW}}$ and $\lambda_{\text{CS}}$, which control the strength of covariate and concept shift modeling. How were these parameters tuned in practice, and how sensitive are the results to their values? It would be helpful to know whether they were selected through validation, grid search, or fixed heuristics, and whether the observed robustness improvements remain stable across different choices of these hyperparameters.

---

> ### Author Response · Authors · 2025-11-28
>
> We thank the reviewer for the thoughtful comments!
>
> ### W1: “unclear how much novel theoretical contribution the analysis provides beyond existing results”
>
> We would like to emphasize two aspects in which our theoretical results are novel. First, the bounds explicitly decompose fairness violation and excess risk into covariate and concept shift components; this decomposition both clarifies the roles of different types of shift and directly motivates the structure of the distribution shift models used in our robust post-processing algorithm.
>
> Second, our excess-risk bound reveals a specific dependence on the fairness tolerance that arises only in the attribute-blind setting, highlighting a robustness disadvantage of attribute-blind models that, to our knowledge, has not been discussed in prior work. In particular, the bound shows that, in the worst case, when the fairness tolerance $\alpha$ is set to small values, even a small fairness violation ($\varepsilon > \alpha$) can induce a huge excess risk $\approx \\|\ell\\|\_\infty$ (recall that the excess risk always is upper bounded by $\\|\ell\\|\_\infty$).
>
> ### W2: “proposed robust post-processing algorithm appears somewhat disconnected from the theoretical results”
>
> We would like to clarify that our robust post-processing algorithm does not require the initial group predictor (that predicts $(A,Z)$ from $X$) to be highly accurate. What is needed is that the true conditional distribution on the target domain lies within the uncertainty set, which we explicitly model as a concept shift relative to the source group predictor. Thus, when the group predictor is estimated with inaccuracies, these discrepancies can be treated and handled as bounded concept shifts within our framework.
>
> ### W3: “experimental evaluation is limited to a single benchmark”
>
> We appreciate the reviewer’s comment and acknowledge the lack of multiple datasets/modalities. However, we believe this is compensated by the fact that we conduct three distinct sets of experiments with different setups: (i) geographic shifts across states, (ii) worst-case covariate shifts, and (iii) group label noise (concept shift). Together, these experiments are designed to systematically probe both components (covariate and concept shift) of our distribution shift models within our robust post-processing algorithm.
>
>
> ### Question: on selecting the regularization hyperparameters $\lambda_{IW}, \lambda_{CS}$ and their sensitivity
>
> In all experiments, the regularization parameters $\lambda_{IW}, \lambda_{CS}$ were tuned on a held-out validation set: we performed a small grid search over a few candidate values and selected the setting that best balanced robustness (fairness under shift) and performance on the validation distribution.
>
> Regarding their sensitivity, we observe the following. First, in the covariate shift and concept shift experiments (Appendices F and G), smaller values of these hyperparameters consistently led to stronger robustness. Second, in the geographic shift experiments, we did not observe substantial performance deviations across alternative hyperparameter settings (see Figure 11, page 30). Overall, we did not find these hyperparameters to induce unstable or inconsistent behavior, although validation is necessary for selecting the right balance between robustness and performance.
>
> Third, regarding calibrating the hyperparameters $\lambda$ to the shift magnitudes, the practitioner can empirically verify whether a specific worst-case covariate shift is covered by running experiments similar to those in Appendix G, which directly evaluates the robustified fair classifier against the worst-case bounded shift (a similar setup can be constructed for concept shift).

---

### Official Review · Reviewer_UfbC · 2025-10-31

**Soundness:** 3
**Presentation:** 3
**Contribution:** 2
**Rating:** 6
**Confidence:** 3

**Summary:**

This paper studies the robustness of group fairness guarantees when machine learning classifiers are deployed under distribution shifts. The authors develop theoretical bounds on both fairness violation and excess risk for randomized fair classifiers, decomposing the effects of covariate and concept shifts. They further show that attribute-aware classifiers (which explicitly include sensitive attributes) enjoy stronger robustness guarantees than attribute-blind ones.

Building on these insights, the authors propose a robust post-processing algorithm—a cutting-set method that alternates between identifying the worst-case distributional perturbation (pessimization) and re-optimizing the classifier to maintain fairness across all discovered perturbations. The algorithm extends the LinearPost framework to multiple distributions and defines the uncertainty set in terms of parameterized covariate and concept shift models.

Empirically, the method is evaluated on the ACSIncome dataset under geographic shifts, showing improved fairness generalization across regions at the cost of reduced accuracy on the training distribution.

**Strengths:**

Overall, the paper is well-written, logically structured, and effectively connects the theoretical analysis with algorithmic design. It provides rigorous and thorough theoretical analysis by providing general, decomposed bounds on fairness violation and excess risk under distribution shift. The cutting-set approach with a pessimization oracle is conceptually clean and theoretically justified. Convergence guarantees are provided, and the method integrates neatly with existing post-processing frameworks.

**Weaknesses:**

The theoretical results, including the techniques and types of bounds, are not particularly surprising, as similar findings exist in prior literature (e.g., Chen et al., 2022), including the conclusion that label shift does not affect fairness. On the practical side, the pessimization step and parameterized shift models introduce several hyperparameters ($\lambda_{IW}$, $\lambda_{CS}$) that may be difficult to tune, and scalability to large datasets or complex shifts is a bit unclear. In addition, the cutting-set iterations could impose significant computational overhead when uncertainty sets are large or fairness constraints are tight.

**Questions:**

1. How sensitive are the results to the hyperparameters defining the uncertainty set ($\lambda_{IW}, \lambda_{CS}$)?
Can practitioners estimate these parameters realistically without labeled target data?

2. How does your approach compare empirically and conceptually to adversarial robustness–based fairness methods (e.g., FR-Train, adversarial reweighting)?

3. The theoretical results are about randomized classifiers; in practice, deterministic deployment is often required. How do the authors envision translating this to deployed models?

---

> ### Author Response · Authors · 2025-11-28
>
> We thank the reviewer for the thoughtful comments!
>
> We would like to address the questions raised by the reviewer, followed by the weaknesses:
>
> ### Q2: compare with adversarial fairness methods.
>
> Our implementation of the pessimization step via gradient ascent with neural networks bears a resemblance to adversarial fairness methods. However, our approach is different in that it is post-processing based and thus applicable to any base predictor (in particular, the GBDT models we use), whereas adversarial training methods typically require end-to-end differentiability. But perhaps more fundamentally:
>
> 1. Theoretically, our iterative cutting-set algorithm provides an explicit guarantee: at convergence, the learned post-processor satisfies the fairness constraints simultaneously for all perturbations discovered by the adversary (implicitly implementing an uncertainty set). By contrast, for robust adversarial fairness methods [1,2], it is generally unclear whether convergence to a min-max saddle point provides such an explicit robustness guarantee with respect to any underlying uncertainty set.
>
> 	Furthermore, our framework allows modeling and robustifying against more general forms of distribution shifts (where adversarial reweighting can be viewed as an instance of covariate shift).
>
> 2. Empirically, while we do not compare directly to robust adversarial fairness methods in this work, the original LinearPost paper [3] already compared post-processing against adversarial fair-representation learning method [4], and observed that adversarial learning struggled to attain high-fairness. We expect similar challenges in the more difficult robustness-under-shifts setting we study.
>
> ### Q3: randomized vs. deterministic classifiers.
>
> This is an important fine point that we also care about, and we propose one practical solution: to obtain deterministic predictions, one can replace explicit randomness with a deterministic hash (e.g., based on a customer ID), and perform randomization seeded with the hash value. One concern is that this may violate individual fairness, since individuals with the same or highly similar inputs may receive different outcomes. However, we note that deterministic neural networks learned via adversarial methods may also fail individual fairness as their decision boundaries can be very jagged.
>
> ### Q1: sensitivity to the hyperparameters $\lambda_{IW},\lambda_{CS}$, and how to estimate them without labeled target data (also related to W2).
>
> We acknowledge the weakness of performing pessimization via gradient ascent on neural nets, which is used in our experiments mainly for simplicity; although we would like to point out that our framework allows for other pessimization methods. For instance, for covariate shift, one idea is to solve for the worst-case sample weight exactly as in Appendix G’s covariate shift experiments, and then use, e.g., Gaussian mixture models to extrapolate (which have more concrete statistical analysis in the literature).
>
> Regarding the sensitivity of the hyperparameters $\lambda_{IW}$ and $\lambda_{CS}$, we observe the following. First, in the covariate shift and concept shift experiments (Appendices F and G), smaller values of these hyperparameters consistently led to stronger robustness. Second, in the geographic shift experiments, we did not observe substantial performance deviations across alternative hyperparameter settings (see Figure 11, page 30). Overall, we did not find these hyperparameters to induce unstable or inconsistent behavior, although validation would be necessary for selecting the right balance between robustness and performance.
>
> ### W1: “The theoretical results… are not particularly surprising, as similar findings exist in prior literature”
>
> We would like to emphasize two aspects in which our theoretical results are novel. First, the bounds explicitly decompose fairness violation and excess risk into covariate and concept shift components; this decomposition both clarifies the roles of different types of shift and directly motivates the structure of the distribution shift models used in our robust post-processing algorithm. Second, our excess-risk bound reveals a specific dependence on the fairness tolerance that arises only in the attribute-blind setting, highlighting a robustness disadvantage of attribute-blind models that, to our knowledge, has not been discussed in prior work.
>
> ---
>
> [1,2] are as in the review
> [3] Xian and Zhao, A Unified Post-Processing Framework for Group Fairness in Classification, 2024.
> [4] Zhang et al., Mitigating Unwanted Biases with Adversarial Learning, 2018.

---

### Official Review · Reviewer_Pie8 · 2025-10-31

**Soundness:** 4
**Presentation:** 4
**Contribution:** 3
**Rating:** 4
**Confidence:** 5

**Summary:**

This paper addresses the problem that group fairness guarantees often fail when a machine learning model is deployed in an environment where the data distribution has shifted from the one it was trained on. The authors first provide a theoretical analysis of how distribution shifts—decomposed into covariate shift and concept shift—affect the fairness and accuracy of randomized classifiers. They derive upper bounds on both the fairness violation and the excess risk under these shifts. Based on this analysis, the paper proposes a new robust post-processing algorithm designed to learn classifiers that maintain fairness guarantees within a defined "uncertainty set" of potential distribution shifts. This algorithm iteratively finds the worst-case distribution shift that maximizes fairness violation and then retrains the classifier to be fair against this new shift. The algorithm is evaluated on the "Retiring Adult" dataset, where it is tested against geographic distribution shifts. Additional results on noisy sensitive attributes and covariate shift are included in the Appendix.

**Strengths:**

- While other works have analyzed fairness under distribution shifts, Theorem 3.1 which decomposed the fairness violation and excess risk bounds into distinct covariate shift and concept shift components is a novel and insightful result.
-  The introduction and preliminaries do an excellent job of motivating the problem with a clear example (Fig. 1) and precisely defining the key concepts, including the fairness criteria and types of distribution shifts. The structure of this paper is easy to follow, and the results are presented clearly. There are rich contents in the Appendix with more theoretical proofs and experimental results.
- The problem of fair classification under distribution shifts is one of the most critical and practical challenges in operationalizing fair ML. This paper provides a tangible, algorithmic solution for this.

**Weaknesses:**

- The proposed post-processing algorithm is an extension of LinearPost (Xian & Zhao, 2024). This reliance limits the work's originality, as the novel solution (Theorem 4.2) is a multi-distribution version of the original Theorem 4.1, and the proof techniques naturally follow the same ideas (as shown in Appendix D.2). More importantly, while the theoretical insights on decomposing shifts (Theorem 3.1) are general and broadly applicable, the proposed solution is narrowly tied to this specific post-processing framework.
- The entire optimization framework hinges on the specification of the uncertainty set $Q$, which is modeled via parameterized neural networks and regularized by a pair of $\lambda_{IW}$ and $\lambda_{CS}$. And the worse-case perturbation $q$ is optimized by gradient ascent on two-layer LeakyReLU network. This seems not making too much sense. On one hand, the distribution shifts are fully controlled by the hyper-parameters $\lambda_{IW}$ and $\lambda_{CS}$. The performance will vary significantly with different settings of $\lambda$s. In a real-world scenario, a practitioner will not have access to multiple target distributions to validate these hyperparameters. On the other hand, the distribution shift itself is modeled by the simple network $q$. This adversarial learning procedure will limit the sensitivity and robustness of the distribution shifts. If the true, real-world distribution shift is more complex than this simple network can represent, the algorithm will only be robust to a small, non-representative subset of shifts, and its fairness guarantees will fail.
- The optimization of Algorithm 1 is problematic as the number of constraints is growing with the number of iterations $T$. The linear program could become very large and computationally expensive to solve.

**Questions:**

1. How tight is the bound of fairness violation in Theorem 3.1?
2. In a real-world scenario where a practitioner has no access to target data, how would you do the model selection regarding different $\alpha$, $\lambda_{IW}$ and $\lambda_{CS}$ values?
3. How did you conclude that Algorithm 1 only need 5 to 20 iterations to converge? If we set $N = 1,000$ samples and $K = 2$ for binary classification, with $\tau = 0.1$ we will need $10^{2000}$ iterations to converge.

---

> ### Author Response · Authors · 2025-11-28
>
> We thank the reviewer for the thoughtful comments!
>
> ### W1: “the proposed solution is narrowly tied to this specific post-processing framework”
>
> We appreciate this concern and would like to clarify that: (1) Algorithm 1 is a meta-algorithm that can be instantiated with any post-processing algorithm that supports multiple distributions; it is not inherently tied to LinearPost. And (2) we chose to instantiate it with LinearPost because it is general: it supports a variety of fairness criteria and problem settings, and extending it to the multi-distribution case was conceptually straightforward given the original formulation. In fact, we believe the strategy we used to extend LinearPost to multiple distributions can also be applied to alternative post-processing algorithms [1,2,3], which could then serve as additional instantiations of Algorithm 1.
>
> ### W2: regarding the uncertainty set.
>
> We interpret the reviewer’s concerns as touching on (i) how the uncertainty set is specified and validated, and (ii) the potential mismatch between its abstract definition and our parameterization using neural nets. We address these in turn.
>
> 1. On validating the uncertainty set (also related to Q2).
>
>     We agree that our robustness guarantee holds only if the actual distribution shift is captured by the chosen uncertainty set. If nothing is known about the test distribution, this cannot be verified a priori. In fact, if the true shift lies largely outside the uncertainty set, the robustification procedure may even worsen fairness on that distribution. We believe this explains why some states in our geographic-shift experiments exhibit regressed fairness (Figure 2): the shifts between those states and CA are plausibly not well captured by the implicitly specified uncertainty set (via $\lambda_{IW}=20$ and $\lambda_{CS}=500$).
>
>     In practice, the practitioners would need to incorporate some information about the target distribution when specifying the uncertainty set. For example, using data from (or close to) the target so that concept shift can be modeled by training group predictors on target labeled data, and covariate shift via importance-weight estimation. Or using knowledge about (bound on) the magnitude of the shift from the source and defining the uncertainty set as all shifts within that radius.
>
> 2. On the mismatch between design of Q and implementation using NN.
>
>     For modeling worse-case bounded shifts, the reviewer raised concerns that (1) they may not be captured by the “simple network” used in our experiments, either due to limited expressivity or optimization difficulties, and (2) it’s hard to calibrate the hyperparameters $\lambda_{IW}, \lambda_{CS}$ to the magnitude of the shifts.
>
>     First, we wish to point out that our framework is not restricted to using neural nets to model the distribution shifts: one could use any parameterized or nonparametric function class as long as (1) it can output group predictions (to simulate concept shift) and represent importance weights, and (2) there is an algorithm to find the worst-case witness in the class.
>
>     Second, from a purely theoretical perspective, by universal approximation theorems, we argue that neural networks should have sufficient representation power to model any worst-case shift, and optimization difficulties (gradient ascent) can be mitigated by increasing network width and entering into the NTK regime [4, 5] (although practitioners should be mindful of generalization properties and sample complexity when performing pessimization). Although the neural nets used in our experiments are simple, they nonetheless demonstrated concrete improvements to robustness across our three sets of experiments, and serve as a good starting point for future explorations on more tailored and sophisticated uncertainty set specifications.
>
>     Third, regarding calibrating the hyperparameters $\lambda$ to the shift magnitudes, while we do not have a theoretical result relating them, because this also depends on the model (architecture) and the optimizer, the practitioner can empirically verify whether a specific worst-case covariate shift is covered by running experiments similar to those in Appendix G, which directly evaluates the robustified fair classifier against the worst-case bounded shift (a similar setup can be constructed for concept shift).
>
> ---
>
> [1] Chen et al., Post-Hoc Bias Scoring is Optimal for Fair Classification, 2024.
> [2] Alghamdi et al., Beyond Adult and COMPAS: Fairness in Multi-Class Prediction, 2022.
> [3] Hardt et al., Equality of Opportunity in Supervised Learning, 2016.
> [4] Ji et al., Neural tangent kernels, transportation mappings, and universal approximation, 2020.
> [5] Allen-Zhu et al., A Convergence Theory for Deep Learning via Over-Parameterization, 2019.

---

> > ### Author Response · Authors · 2025-11-28
> >
> > ### Q3 and W3: number of iterations of Algorithm 1, and “linear program could become very large and computationally expensive to solve”
> >
> > We thank the reviewer for highlighting the lack of clarity in our statement that “Algorithm 1 only needs 5-20 iterations to converge.” What we intended to convey is that in our empirical implementation, using simple neural nets and gradient ascent for pessimization, the cutting-set procedure typically fails to find additional violating witnesses after 5-20 iterations (despite we do 3 retries, it is possible that we may have terminated early due to optimization difficulties). We will rephrase the text to make this explicit.
> >
> > Regarding the theoretical convergence bound, it is a worst-case guarantee and, as is common with cutting-set and robust optimization methods, is likely to be pessimistic in practice.
> >
> > We acknowledge that the size of the linear program (LP) in our algorithm grows with T, which is an inherent limitation of cutting-set methods. However, for the range of T we consider and the size of our evaluation datasets (around 20,000 training examples), (i) the LP is consistently solved within about one minute, and (ii) modern LP solvers provide support for scaling to larger instances. For more demanding cases, one could also resort to approximate or stochastic optimization over the LP with warm start; so we may view this as a practical engineering trade-off rather than a fundamental barrier.
> >
> > ### Q1: “How tight is the bound of fairness violation in Theorem 3.1?”
> >
> > The bound in Theorem 3.1 is tight in the same sense as Theorem 3.2: one can construct examples (similar in spirit to the discrete Example 1) for which the fairness-violation bound is achieved up to multiplicative constants.

---

### Official Review · Reviewer_MXN8 · 2025-11-01

**Soundness:** 3
**Presentation:** 3
**Contribution:** 3
**Rating:** 6
**Confidence:** 4

**Summary:**

The paper addresses the core issue of robustness in group fairness guarantees under distribution shifts. It recognizes that statistical fairness of a model's outputs on a data distribution can be easily invalidated when a classifier trained to be fair on a source distribution is used in a new target distribution. This is empirically demonstrated with an income prediction task where a classifier fair on data from California becomes increasingly unfair and less accurate when evaluated on other states, correlating with the magnitude of the distribution shift. Then the work provides a theoretical analysis of this problem and an robust algorithm derived from this analysis.

**Strengths:**

1. This paper's primary strength lies in its deep theoretical analysis. The decomposition of fairness violation into covariate and concept shifts provides a clear and actionable framework. The discovery of the dependency in the excess risk bound for attribute-blind classifiers is also an important contribution. The Reviewer did not find major incorrectness within the theoretical proof.
2. The usage of the proposed method combined with post-processing is considered to be a reasonable approach. The extension of LinearPost to the multi-distribution setting is a technical contribution. The uncertainty set is then defined via parameterized models, the algorithm is general and can be adapted to various shift scenarios.
3. The proposed algorithm is directly motivated by the theoretical findings. The construction of the uncertainty set $Q$ by parameterizing covariate and concept shifts directly mirrors the decomposition in the fairness violation bound (Theorem 3.1). This connection between analysis and method makes the overall work compelling.

**Weaknesses:**

1. The core of the robust algorithm is in the **pessimization** phase, which finds the worst-case distribution by optimizing the parameters of neural networks via gradient ascent. This is a non-convex optimization problem nested within the main loop, and the authors acknowledge that this step may not be performed exactly and that its performance can be variable. This introduces potential instability and makes the algorithm highly sensitive to the choice of regularization hyperparameters and optimization details (learning rate, number of steps), which may be difficult to tune in practice.
2. Following the above question, the bilevel min-max optimization framework to optimize the worst-case performance sounds not novel, such as re-weighting including but not limited to distributionally robust optimization, see [1,2,3,4]. Could the authors please distinguish the proposed framework with other worst-case optimization methods?
3. The overall algorithm appears computationally intensive, which involves an outer loop adding new constraints, and an inner loop for the adversarial training of neural networks. While the paper notes that the algorithm typically converges in 5-20 iterations, it does not provide a direct comparison of computational costs against the baselines. This computational overhead could be a barrier to applying the method to large datasets or large scale networks, limiting the reproducibility.

[1] Online fairness-aware learning under class imbalance, ICML 2020.
[2] FIFA: Making Fairness More Generalizable in Classifiers Trained on Imbalanced Data, ICLR 2023.
[3] Fairness-aware class imbalanced learning on multiple subgroups, UAI 2023.
[4] On the Inductive Biases of Demographic Parity-based Fair Learning Algorithms, UAI 2024.

**Questions:**

Please see above weaknesses.

---

> ### Author Response · Authors · 2025-11-28
>
> We thank the reviewer for the thoughtful comments!
>
> ### W1: pessimization via gradient ascent on neural nets introduces potential instability and sensitivity to hyperparameters
>
> We acknowledge the weakness of performing pessimization via gradient ascent on neural nets, which is used in our experiments mainly for simplicity; although we would like to point out that our framework allows for other pessimization methods. For instance, for covariate shift, one idea is to solve for the worst-case sample weight exactly as in Appendix G’s covariate shift experiments, and then use, e.g., Gaussian mixture models to extrapolate (which have more concrete statistical analysis in the literature).
>
> Regarding the sensitivity of the hyperparameters $\lambda_{IW}$ and $\lambda_{CS}$, we observe the following. First, in the covariate shift and concept shift experiments (Appendices F and G), smaller values of these hyperparameters consistently led to stronger robustness. Second, in the geographic shift experiments, we did not observe substantial performance deviations across alternative hyperparameter settings (see Figure 11, page 30). Overall, we did not find these hyperparameters to induce unstable or inconsistent behavior, although validation would be necessary for selecting the right balance between robustness and performance.
>
> ### W2: bilevel min-max optimization framework vs. adversarial fairness methods
>
> Our implementation of the pessimization step via gradient ascent with neural networks bears a resemblance to adversarial fairness methods. However, our approach is different in that it is post-processing based and thus applicable to any base predictor (in particular, the GBDT models we use), whereas adversarial training methods typically require end-to-end differentiability. But perhaps more fundamentally:
>
> Theoretically, our iterative cutting-set algorithm provides an explicit guarantee: at convergence, the learned post-processor satisfies the fairness constraints simultaneously for all perturbations discovered by the adversary (implicitly implementing an uncertainty set). By contrast, for robust adversarial fairness methods [1,2,3,4], it is generally unclear whether convergence to a min-max saddle point provides such an explicit robustness guarantee with respect to any underlying uncertainty set.
> Furthermore, our framework allows modeling and robustifying against more general forms of distribution shifts (where adversarial reweighting can be viewed as an instance of covariate shift).
> Empirically, while we do not compare directly to robust adversarial fairness methods in this work, the original LinearPost paper [5] already compared post-processing against adversarial fair-representation learning method [6], and observed that adversarial learning struggled to attain high-fairness. We expect similar challenges in the more difficult robustness-under-shifts setting we study.
>
> ### W3: the overall algorithm appears computationally intensive
>
> We acknowledge that the size of the linear program (LP) in our algorithm grows with T, which is an inherent limitation of cutting-set methods. However, for the range of T we consider and the size of our evaluation datasets (around 20,000 training examples), (i) the LP is consistently solved within about one minute, and (ii) modern LP solvers provide support for scaling to larger instances. For more demanding cases, one could also resort to approximate or stochastic optimization over the LP with warm start; so we may view this as a practical engineering trade-off rather than a fundamental barrier.
>
> ---
>
> [1-4] are as in the review
> [5] Xian and Zhao, A Unified Post-Processing Framework for Group Fairness in Classification, 2024.
> [6] Zhang et al., Mitigating Unwanted Biases with Adversarial Learning, 2018.

---

### Meta-Review · Area_Chair_LtKi · 2026-01-06

**Summary:**

This paper studies the robustness of group fairness guarantees under distribution shifts. Its main strength lies in the theoretical analysis, which decomposes fairness violation and excess risk into covariate and concept shifts and provides new insights into the robustness disadvantages of attribute-blind classifiers. Building on this analysis, the authors propose a robust post-processing algorithm to address group fairness under distribution shift.

While reviewers acknowledge the theoretical analysis, there are concerns regarding the paper’s practical applicability, particularly its reliance on a heavily parameterized uncertainty set that may be difficult to specify or validate without access to target data. The proposed pessimization procedure further raises concerns about hyperparameter sensitivity, scalability, and robustness to real-world shifts. Although the rebuttal clarifies several points, core concerns about practical usability, empirical generality, and limited algorithmic novelty remain unresolved. I therefore recommend rejecting the paper in its current form.

**Reviewer Concerns:**

The rebuttal addressed several clarification questions, including the theoretical positioning of the work, conceptual difference from adversarial fairness methods, and explanations of algorithmic design choices and convergence behavior.

However, core concerns regarding the limited algorithmic novelty over prior work, the practical validity of the uncertainty set without access to target data, the scalability of the pessimization procedure, and the empirical generality of the proposed method (using more datasets and direct empirical comparison with adversarial fairness methods) were not fully addressed.

**Reviewer Scores:**

Since concerns about practical applicability, limited empirical scope, and incremental contribution were shared across reviewers and not fully resolved in the rebuttal, I do not expect their scores to change substantially after discussion.

---

### Decision · Program_Chairs · 2026-01-26

Reject